# A temperature-adaptive component-dynamic-coordinated strategy for high-performance elastic conductive fibers

Yue Zhang[1], Zechang Ming[2], Zijie Zhou[1], Xiaojie Wei[2], Jingjing Huang[2], Yufan Zhang[3], Weikang Li[1], Liming Zhu[1], Shuang Wang[1], Mengjie Wu[1], Zeren Lu[2], Xinran Zhou [3] & Jiaqing Xiong [1,3] ✉

Temperature-adaptive elastic conductive fibers (ECFs) are crucial for seamlessly integrating electronic textiles, promoting the development of wearables, soft robotics, and high/low-temperature electronics. Realizing ECFs with balanced elasticity, conductivity, and temperature adaptivity remains challenging due to the difficulty of coupling the mechano-electrical-thermal properties at a microscale fiber. We design a wet-spun ECF consisting of thermoplastic polyurethane (TPU), silver flakes (AgFKs) and liquid metal microspheres (LMMSs) with regularly arranged filler architecture, revealing a cold/thermal stretching activated tricomponent-dynamic-coordination mechanism for autonomously-enhanced electrical conductivity (from ~1070 S cm$^{-1}$ at 25 °C to 1160 S cm$^{-1}$ at −30 °C and 3020 S cm$^{-1}$ at 180 °C) and improved electrical stability to sustain 1000 stretching cycles (60% strain at 80 °C). The fiber exhibits scalability and favorable knittability, demonstrating e-textiles such as biomedical electrodes, high/low-temperature near-field communication gloves, and intelligent firefighting suits. The autonomous mechano-thermo-electrical coupling strategy can inspire high-performance and environment-adaptive ECFs for extreme applications.

Electronic textiles (e-textiles) require non-obtrusive integration of functional modules to break through the impact of rigid and bulky electrical modules on the mechanical compliance of traditional electronics[1–3]. The integration capability and functionality of electronic modules are highly dependent on the properties of conductive components, especially under extreme mechanical and thermal conditions[4,5]. Conductive fiber is one of the most promising electrode material for e-textiles, because of its seamless integration capability that ensures electrical stability and wearable comfort[6].

Traditional conductive fibers generally consist of carbon fiber, ionic hydrogel, or metallic wires, each of which has distinct performance superiorities and limitations determined by the composition. Carbon fibers or yarns have high-temperature tolerance but possess

moderate conductivity ($10^2$–$10^5$ S cm$^{-1}$) and very low elasticity (~1–2%)[7]. Elastic hydrogel fibers with ionic conductivity have been widely exploited in recent years. Yet, their practical potential is restricted due to the challenges in electrical conductivity (~400 S cm$^{-1}$) and thermal stability[8–11]. Metallic fibers such as stainless steel, copper, and silver possess high conductivity and thermal stability but are not impressive in terms of elasticity and stitchability[12,13]. Realizing a trade-off between conductivity, elasticity, stitchability, and temperature adaptivity requires strategies to reconcile the superiorities of metallic conductivity and polymer deformability, paving the way for efficient processing of e-textiles with elevated comfort and environmental adaptivity[14].

Elastic conductive fibers (ECFs) are an emerging type of conductive fibers with stretchability, which are highly promising in wearable

[1]State Key Laboratory of Advanced Fiber Materials, College of Textiles, Donghua University, Shanghai, China. [2]College of Materials Science and Engineering, Donghua University, Shanghai, China. [3]Innovation Center for Textile Science and Technology, Donghua University, Shanghai, China. ✉e-mail: jqxiong@dhu.edu.cn

electronics due to deformation compliance. ECFs commonly contain organic elastomer matrixes with inorganic conductive fillers to balance the conductivity and elasticity[15,16], such as fibers composed of matrices of polyurethane (PU)[17], poly(styrene-butadiene-styrene) (SBS)[18], poly(styrene-ethylene-butylene-styrene) (SEBS)[19], rubber[20], and fillers of metal micro-nano wires, particles or flakes[21,22], liquid metal (LM)[23], carbon materials[24], and MXene[25], etc. So far, a state-of-the-art ECF consisting of fluoro-elastomer and LM achieved a conductivity of 435 S cm$^{-1}$ and a fracture strain of 1170%[26]. An ECF consisting of PU and poly(3,4-ethylenedioxythiophene)-poly(styrenesulfonate)/polyvinyl alcohol (PEDOT:PSS/PVA) achieves a conductivity of 147 S cm$^{-1}$ and a fracture strain of 500% for stretchable electrodes in a wristband[27]. These pioneering works inspired various recipes to boost the electromechanical performances of ECFs. However, limited attempts were dedicated to exploiting ECFs' electromechanical robustness at high or low-temperature conditions, which is significant for extending the application scenario and achieving environment-adaptive e-textiles. Due to limitations in material properties, fabrication strategies and fiber geometries, there still lacks an ECF that can balance conductivity, elasticity, stitchability, and temperature tolerance, to maintain performances in weaving or knitting that require large deformations, and adapt to cryogenic and high temperature scenarios[28].

In this work, an ECF was developed via wet spinning using a recipe of thermoplastic polyurethane (TPU), silver flakes (AgFKs), and eutectic gallium-indium liquid metal microspheres (LMMSs of EGaIn), named PUAL fiber. The wet-spinning process provides shear and traction forces to improve the fiber's alignment in terms of the TPU polymer chains and the AgFKs' arrangement (Fig. 1a). With the mature recipe and facile wet-spinning process, the PUAL fiber can be produced on a large scale, reaching a length of 60 m, with a uniform diameter, stable mechanical properties (fracture strain of 450%, ultimate strength of 10.6 MPa) and consistent conductivity (~1070 S cm$^{-1}$) (Fig. 1b, c, Supplementary Table 1 and Supplementary Note 1). A fiber with a diameter of ~350 μm exhibits high deformability, allowing it to be knotted or knitted into 3D structures (Fig. 1d). Through a tricomponent-dynamic-coordination mechanism, the fiber can autonomously enhance its electrical conductivity under high (heating to 180 °C) or low temperatures (freezing to −30 °C) (Fig. 1e), and improve its electrical stability by cyclic stretching activation at the same temperature range. This PUAL fiber shows mechano-thermo-electrical cooperative superiorities compared with other ECFs[29–35] (Fig. 1f and Supplementary Table 2), promising for temperature-adaptive e-textiles, human-machine interaction, and environment exploration and wireless communications, etc.

## Results

### Design and fabrication of PUAL fiber with thermal-enhanced conductivity

Figure 2a reveals the cross-section of a representative wet-spun PUAL fiber, which consists of TPU, AgFKs, and LMMSs. As a low-cost commercial elastomer with high adaptability to wet-spinning, TPU is selected as the elastic matrix to protect the metallic fillers. The main conductive filler for forming conductive paths is AgFK (1–5 μm, Supplementary Fig. 1), which shows acceptable cost, high conductivity, and high compatibility with the TPU matrix (Supplementary Fig. 2). With the wet-spinning-induced orientation, AgFKs are capable of forming reliable conductive paths in a fiber. In addition to AgFKs, LMMS is introduced as a soft conductive filler to bridge the AgFKs to construct a binary conductive architecture, which can rupture to release LM for electrical self-compensation under thermal/mechanical stimulations.

The TPU matrix shows favorable thermodynamic behavior, guaranteeing the electromechanical properties of the PUAL fiber. First, no crystal peaks and melting peaks are observed in the X-ray diffraction (XRD) results of the TPU (Supplementary Fig. 3a), verifying an amorphous structure that is advantageous for the chain segment to migrate under heating[36]. Dynamic thermo-mechanical analysis (DMA) reveals a glass transition temperature of −34 °C[37], which is the initial temperature of the rubbery state of the TPU (Fig. 2b). The obvious vector (q) scattering peak appears in the small angle X-ray scattering (SAXS) curve and the two-step weight loss behavior existing in thermogravimetric analysis (TGA) curve suggest that the TPU has a microphase separation feature (Supplementary Fig. 3b, c)[38], rendering a relatively high softening temperature (142 °C) that is verified by thermomechanical analysis (TMA) (Fig. 2c). Besides, the rheological behavior reveals that the TPU chain segment starts to enter a viscous flow state at 182 °C (Supplementary Fig. 3d)[39]. These results indicate that the TPU matrix will undergo reversible volume expansion/contraction between around −30 °C and 180 °C, defining the working temperature range of the PUAL fiber, which lays the foundation of thermal-responsive electrical properties of our ECF.

AgFKs and LMMSs are both crucial to constructing a binary conductive filler to enhance the fiber's conductivity (Fig. 2d, e). To study the specific role of each component, first, a PUL fiber with a high mass ratio of 7 (LMMSs to TPU) (without AgFKs) is confirmed non-conductive due to the isolation of LMMSs (see details in Supplementary Fig. 4). In comparison, PUA (TPU/AgFKs) fibers with the optimal mass ratio of 3.5 (AgFKs to TPU) are conductive, achieving a conductivity of 80 S cm$^{-1}$, with ~530% tensile strain and 11.4 MPa ultimate strength (Fig. 2d and Supplementary Fig. 5). This indicates that AgFK makes a major contribution to forming the conductive path, instead of the LMMSs. Incorporating LMMSs with median particle sizes of 20 μm (PUAL$_{20}$) or 40 μm (PUAL$_{40}$) (Supplementary Note 2) with twice the mass of the AgFKs in the PUA fiber (LMMS:AgFK:PUA = 7:3.5:1) results in optimal PUAL$_{20}$ and PUAL$_{40}$ fibers that show significantly enhanced electrical conductivity to ~890 S cm$^{-1}$ and ~1070 S cm$^{-1}$ respectively (Fig. 2d, the PUAL$_{40}$ fiber with superior conductivity is referred to as PUAL fiber for this study, see details of LMMS content and size effects in Supplementary Figs. 6, 7). The results demonstrate that a cooperative effect exists between the AgFKs and LMMSs, where the AgFKs construct a dominant conductive path that can anchor the LMMSs by alloying effect[40,41], and the LMMSs provide electrical bridges to enhance the AgFKs' connection, both cooperatively promoting the fiber's conductivity.

The wet-spinning process for PUAL fiber fabrication also plays an important role in the fiber's properties. Wet-spinning is an industrial high-throughput fiber fabrication technique that creates continuous fibers by extruding a spinning solution through a needle into a coagulation bath, where the fiber experiences shear flow at the extrusion needle and axial traction forces out of the needle[42,43]. Based on this recipe, the shear and traction forces during wet-spinning induce the rotation of the two-dimensional AgFKs to form a high orientation (this does not influence the zero-dimensional LMMSs). This high orientation of the AgFKs along the fiber's axial direction can be observed in the longitudinal-section and cross-section morphologies of the PUA and PUAL fibers (Fig. 2e and Supplementary Figs. 8, 9). In comparison, cast films with the same compositions show a random filler distribution (Fig. 2e, Supplementary Figs. 8, 9 and Supplementary Note 3). As illustrated in Fig. 2f, the orientation ratio of the AgFKs in the fibers can be calculated by counting the number of AgFKs with an angle (θ) less than 45 degrees with the fiber's axial direction (it is not suitable for the film samples since they do not have an axial direction and include significant numbers of out-of-plane-oriented AgFKs). Both the representative fiber samples with and without LMMSs (PUA and PUAL) are found to achieve high AgFK's orientation over 80% (Supplementary Fig. 9), which is also complementally confirmed by the strong angle-dependent pattern of wide-angle x-ray scattering (WAXS) (see details in Supplementary Fig. 10). The AgFKs' orientation can be tuned by varying the wet-spinning speed (5–60 mL h$^{-1}$), which influences the

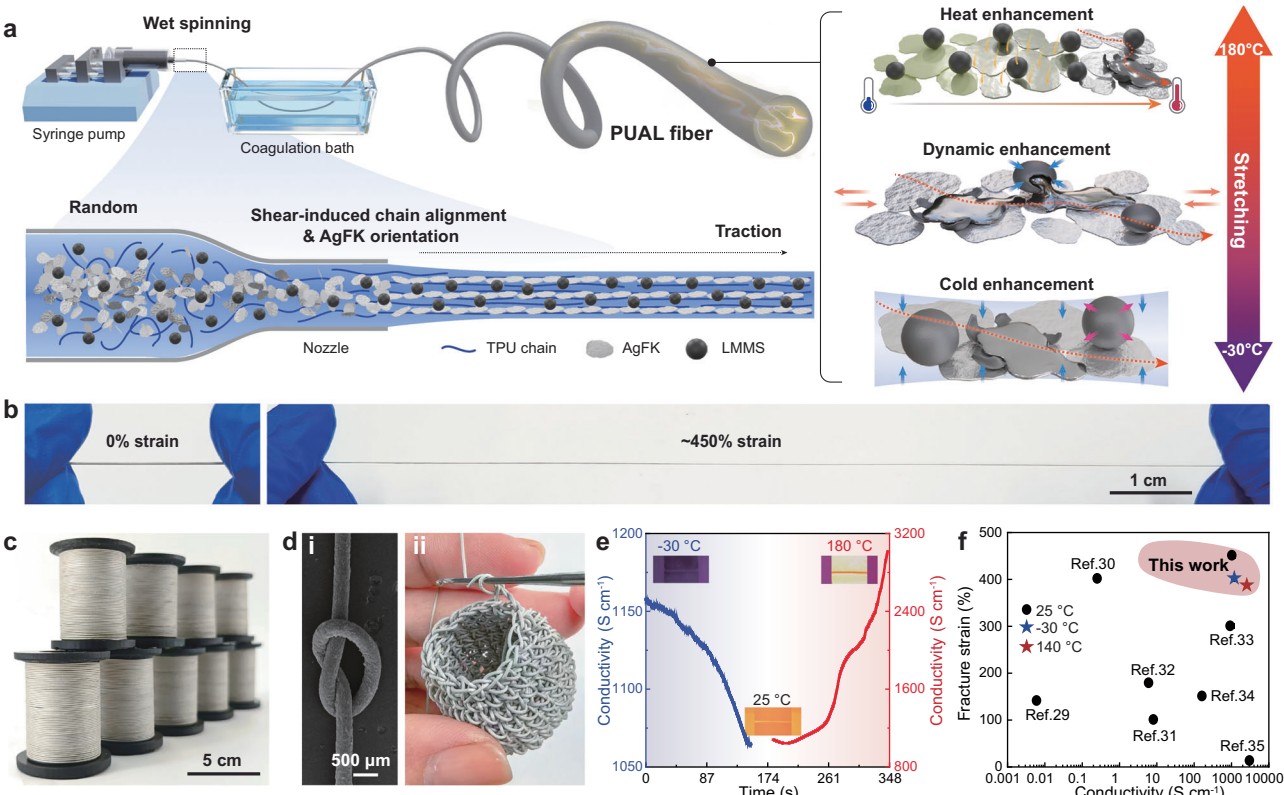

**Fig. 1 | Design and properties of thermoplastic polyurethane (TPU)/silver flake (AgFK)/liquid metal microsphere (LMMS) (PUAL) fiber. a** Schematic illustration of the wet-spun PUAL fiber with enhanced conductivity and electromechanical stability in a wide temperature range. **b** Digital photos showing the stretchability of the PUAL fiber. **c** Digital photo showing the scalability of the PUAL fiber.

**d** Demonstration of (**i**) a knotted PUAL fiber and (**ii**) 3D knitting of PUAL fiber. **e** Conductivity enhancement of the PUAL fiber from 25 °C to −30 °C, and to 180 °C. **f** Electrical conductivity, stretchability, and thermal stability trade-off of the PUAL fiber with other ECFs.

shear stress on the AgFKs, as demonstrated by hydrodynamic finite element analysis and SEM images of AgFKs in the PUAL fiber (Supplementary Figs. 11, 12). An appropriate stress difference between vertical and parallel directions of AgFKs is crucial to rotate AgFKs and promote high orientation, and 20 mL h$^{-1}$ was found to be an optimal wet-spinning speed that can achieve a high orientation of 84% (Fig. 2g). The high shear force and orientation render more AgFKs overlapping to increase stress transmission and electrical connection, enhancing both the fiber's mechanical properties (Fig. 2h) and electrical conductivity (Fig. 2i and Supplementary Fig. 13).

## Static conductivity enhancement of PUAL fiber at high temperatures

To evaluate the PUAL fiber's thermal adaptivity and the tricomponent recipes' cooperative behavior at high temperatures (Fig. 3a), we heated a series of samples (PUL, PUA, PUAL fibers, and the cast films with the same composition as the PUAL fibers) from 25 °C to 140 °C and 180 °C and monitored their conductivity variation (Fig. 3b, c and Supplementary Fig. 14). All samples show higher conductivity at higher temperature, among which the PUAL fiber has highest conductivity at all temperatures, which increases from ~1070 S cm$^{-1}$ to 2025 S cm$^{-1}$, and 3020 S cm$^{-1}$ at 25 °C, 140 °C, and 180 °C, respectively. These results indicate that both AgFK and LMMS function to construct the conductive path at room temperature and elevated temperatures (Supplementary Fig. 13). Wherein, the function of AgFKs' orientation on the conductivity enhancement can be evidenced by the low conductivity of the cast film (increasing from ~88 S cm$^{-1}$ to 115 S cm$^{-1}$, and 124 S cm$^{-1}$ at 25 °C, 140 °C, and 180 °C, respectively) because of its random filler distribution (Fig. 3b, c).

The conductivity enhancement of PUAL fiber at high temperatures can be attributed to the cooperative thermal-induced electrical-connection enhancement mechanism between the components, including the AgFK degreasing, AgFK orientation, and LMMS soft bridging (Fig. 3a and Supplementary Fig. 13). To verify the AgFK degreasing mechanism, the PUAL fiber was heated at different temperatures (25 °C, 100 °C, and 180 °C) for 30 min, during which the lubricants on the AgFKs can be partially removed, showing a surface with increasing roughness and clearer profile (Fig. 3d)[44], and enhancing the AgFKs exposure and electrical connection. This was also evidenced by the significantly diminished Fourier transform infrared spectroscopy (FTIR) peak of hydroxyl groups (-OH) regarding AgFKs in the heated fiber (Fig. 3e). Besides, heating-induced deformation and rupture of LMMSs can release liquid metal (Fig. 3f and Supplementary Fig. 15a)[45,46], which compensates the conductive pathways constructed by the AgFKs, further enhancing the fiber's conductivity. This electrical compensation effect of LMMS is found to mainly function when the temperature reaches up to 150 °C (see details in Supplementary Fig. 4c). The irreversible thermal-induced morphology transformation of the roughened AgFKs and the ruptured LMMSs upon cooling from 180 °C to 25 °C indicates a reliable heating-induced conductivity enhancement mechanism via improving the fillers' electrical connection (Supplementary Fig. 15b).

The fiber exhibits more vigorous TPU segmental chain motion upon heating below 142 °C, where the AgFK degreasing and LMMS deformation/rupture occur gradually. This process can be verified by observing the fiber's conductivity variation when periodically heating it from 25 °C to 100 °C (Fig. 3g). Taking PUAL fiber as an example (Fig. 3g), in the first cycle, the fiber expands, accompanied by thermal

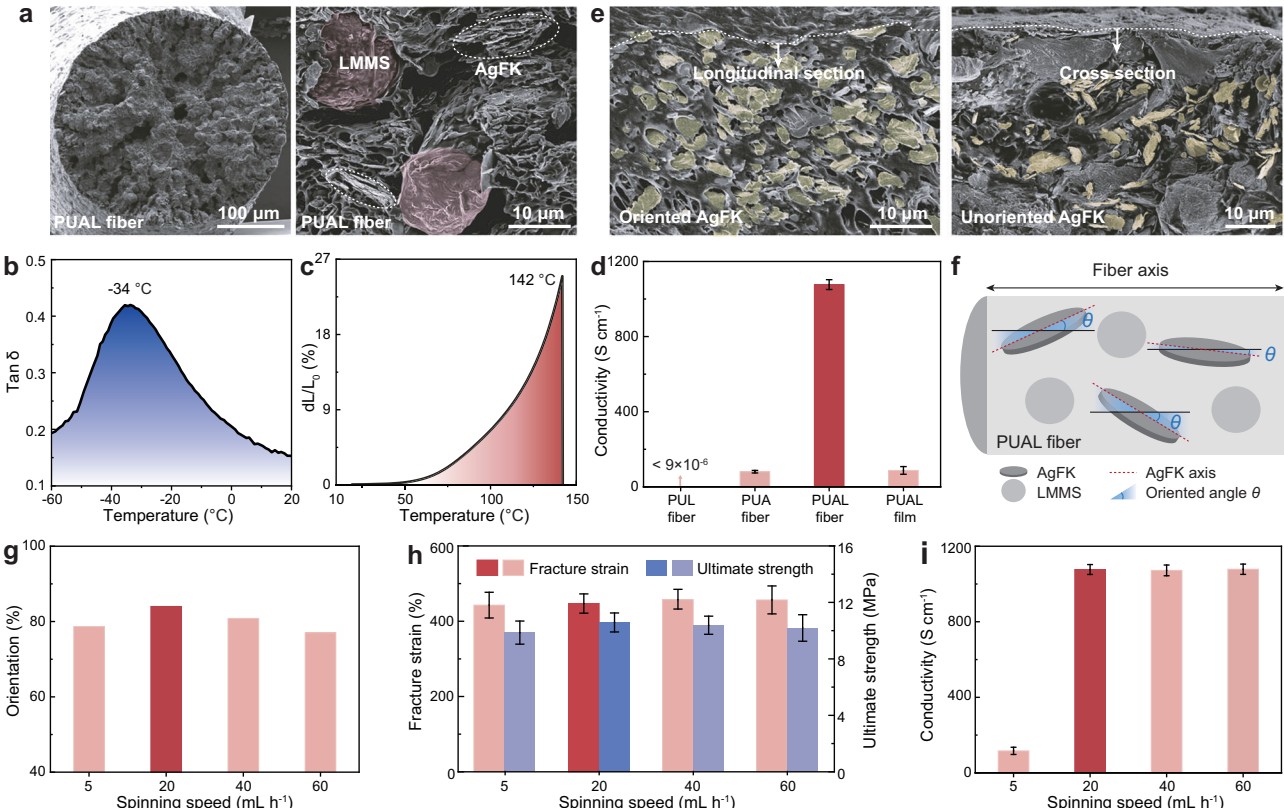

**Fig. 2 | Component-coordination mechanism of PUAL fiber. a** Scanning electron microscope (SEM) image of the cross-section of a representative PUAL fiber. The LMMSs in the fiber are marked in red and AgFKs are marked by white dashed circles. **b** Temperature-dependent tan δ of the PUAL fiber. **c** TMA curve reveals the softening behavior of the PUAL fiber. **d** Conductivity of TPU/LMMS (PUL), TPU/AgFK (PUA), PUAL fibers, and cast films with the same compositions as the PUAL fibers. **e** SEM images of the longitudinal section of the PUAL fiber and cross-section of the PUAL cast film. The main AgFKs are marked in bright yellow. **f** Schematic to illustrate the statistical calculation of AgFKs' orientation degree in the fiber.

**g** AgFKs' orientation degree in the fibers fabricated with different wet-spinning speeds (about 50 AgFKs were randomly taken from PUAL fiber's longitudinal-section SEM images to measure the angle between their long axis direction and the fiber's axial direction, where angles less than 45 degrees were counted as oriented and the ratio of oriented AgFKs was calculated). **h**, **i** Mechanical (**h**) and electrical (**i**) properties of the fibers fabricated with different wet-spinning speeds. Five parallel samples were measured, and error bars represent the standard deviation for (**d**, **h**, **i**).

exfoliation of the lubricants from the AgFKs to expose the AgFKs, which causes a significant increase in the fiber's conductivity from ~1070 S cm⁻¹ to 1361 S cm⁻¹. Upon cooling down to 25 °C, continuous AgFKs degreasing happens, accompanied by contraction of the fiber to the initial volume, promoting the conductive fillers to rearrange and form closer connections, resulting in a further increased conductivity up to 1651 S cm⁻¹ (the starting point at the second cycle). For more heating cycles, despite the reversible conductivity changes caused by TPU thermal expansion and contraction, the fiber's conductivity gradually increased from 1361 S cm⁻¹ to 1431 S cm⁻¹, 1451 S cm⁻¹, 1471 S cm⁻¹, and finally 1470 S cm⁻¹ at the highest temperature (100 °C) of each heating cycle. This increase occurred due to further degreasing and exposure of AgFKs, along with minor electrical compensation from LMMS deformation and rupture. The same trend is observed in the PUA fiber (Supplementary Fig. 16), indicating that the heating-induced AgFK exposure dominates the fiber's conductivity enhancement, especially in the first heating cycle. Thereafter, continuous thermal degreasing of the AgFKs leads to further conductivity enhancement until saturation, as verified by extended multi-cycle heating tests (from 25 °C to 100 °C) over a week with one heating cycle per day (Supplementary Fig. 17).

Therefore, as illustrated in Supplementary Fig. 13, we conclude that AgFK degreasing dominates in improving the fibers' conductivity at high temperatures, and the TPU matrix's reversible volume change governs the conductivity increase during the cooling process. LMMSs

play a tiny role in conductivity improvement during static heating below 100 °C because they mainly rupture when the temperature reaches up to 150 °C. Similar phenomena of thermal-induced conductivity enhancement were found in fibers based on other elastomers (such as SEBS and styrene-isoprene-styrene (SIS)) (Supplementary Figs. 18, 19 and Supplementary Note 4), indicating that the recipe of PUAL fiber represents a universal mechanism of heating-induced autonomous conductivity enhancement.

## Dynamic electromechanical stability enhancement of PUAL fiber at high temperatures

E-textiles inevitably encounter various deformations (intrinsically local tensile deformation) in practical applications, influencing the electromechanical stability of ECFs during cyclic stretching. Apart from the autonomous conductivity enhancement during static heating, our PUAL fiber shows an additional advantage in improving electrical stability under cyclic stretching activation or stretching activation at high temperatures, through dynamically compensated electrical connections between the AgFKs and LMMSs (Fig. 4a). To study this dynamic electromechanical behavior, the PUA and PUAL fibers were subjected to cyclic stretching with 20 cycles at different tensile strains (10%–60%) at 25 °C. The PUAL fiber with lower resistance variation ($\Delta R/R_0$) shows higher electrical stability than the PUA fiber (Fig. 4b, c and Supplementary Fig. 20), indicating that the LMMSs as electrical bridges can be activated by cyclic stretching to release LM and

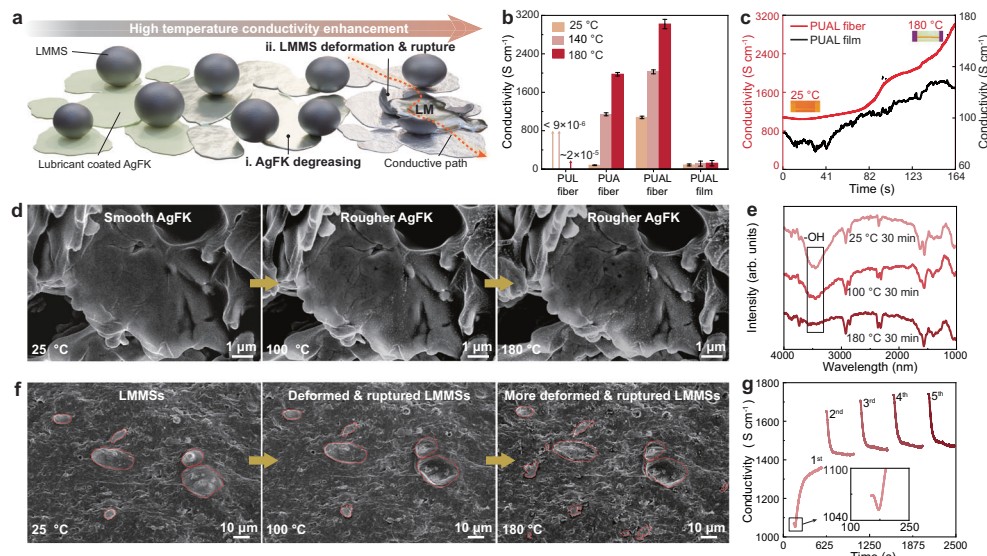

**Fig. 3 | Static conductivity enhancement mechanism of PUAL fiber at high temperatures. a** Schematic diagram of the fiber's heating-induced conductivity enhancement mechanism. **b** Conductivity of PUL, PUA, PUAL fibers, and cast films with the same composition as the PUAL fiber, at 25 °C, 140 °C, and 180 °C (five parallel samples were measured, and error bars represent the standard deviation). **c** Continuous conductivity change of PUAL fiber and PUAL cast film from 25 °C to 180 °C. **d** In situ SEM images of AgFKs in the PUAL fiber at different temperatures. **e** Temperature-dependent FTIR spectra of the AgFKs in the PUAL fiber. **f** In situ SEM images of the LMMSs in the PUAL fiber at different temperatures, as marked by red circles. **g** Conductivity variation of the PUAL fiber when cyclically heated from 25 °C to 100 °C with natural cooling (the start and end points of each curve correspond to the fiber's conductivity at 25 °C and 100 °C, respectively, with a two-hour interval between each cycle).

dynamically repair the electrical connections between the separated AgFKs. The effectively activated electrical paths between AgFKs and LMMSs improve the fiber's electromechanical stability. The same stretching activation effect is also observed in the PUAL₂₀ fiber with smaller LMMSs (20 μm) to a slightly smaller extent than the PUAL fiber, indicating that larger LMMSs provide more reliable electrical stability for the fiber during cyclic stretching (see details in Supplementary Fig. 21).

To study the fiber's electrical stability by considering both mechanical and thermal properties, the PUAL fiber was subjected to cyclic stretching with increasing strains (60–180% strain) at elevated temperatures (60 °C, 80 °C, and 100 °C) to monitor its electrical resistance evolution. As shown in Fig. 4c–e and Supplementary Fig. 22, at room temperature (25 °C), the PUAL fiber shows a significant resistance variation increase ($\Delta R/R_0$ around 500%) at the end of cyclic stretching with 60% strain. In comparison, it delivers significantly improved electrical stability under cyclic stretching between 60 °C and 100 °C, which shows a lower resistance variation ($\Delta R/R_0$ around 160%) even with a much higher tensile strain up to 180% at 80 °C. This demonstrates that thermal-stretching activation can further improve the fiber's electrical stability, due to the cooperative enhancement mechanisms of LMMSs' rupture and AgFKs' degreasing (Fig. 4a and Supplementary Fig. 13). The sufficiently exposed AgFKs can be well bridged by the released LM to achieve robust electrical connections, which highly maintains the fiber's electrical properties to resist higher tensile deformation[41,47].

Considering both electrical and mechanical stabilities at elevated temperatures, the PUAL fiber activated at 80 °C shows better electromechanical stability compared to the samples activated at 60 °C and 100 °C (Fig. 4d and Supplementary Figs. 22, 23). This is because the thermal-stretching activation helps to expose the AgFKs and activate the LMMSs to improve the fiber's electrical stability (Fig. 4e)[48], but also increases the molecular mobility and elastomer relaxation to sacrifice the fiber's mechanical properties (Fig. 4f–h). As revealed in Fig. 4f, the fiber stretched cyclically at 80 °C shows superior elastic recovery than those activated at 60 °C and 100 °C, because comparing with 60 °C,

the higher thermal energy at 80 °C sufficiently promotes the orientation and homogenization of AgFKs and LMMSs, which not only improves the fillers' connection to ensure higher fiber electrical stability but also contributes to higher mechanical properties due to the uniform stress distribution. Meanwhile, 100 °C will overly soften the TPU matrix, making it less elastic to support the fillers and maintain effective mechanical and electrical recovery for the fiber.

Mechanical durability of ECFs remains a huge challenge because ultrafine fibers struggle to withstand stress-induced electrical damage, especially after exposure to extreme thermal and mechanical stimuli. Herein, even with thermal treatment at 80 °C for 30 min, the PUAL fiber still maintains favorable mechanical elasticity with an ultimate strength of 8.2 MPa and a fracture strain of 450%, comparable to that in the ambient condition (10.6 MPa, 450%) (Supplementary Fig. 24). After experiencing 100 stretching cycles with 100% strain under 80 °C, the fiber maintains favorable mechanical stability (Fig. 4g). Furthermore, the fiber can sustain 1000 stretching cycles with 60% strain at 80 °C with a resistance change rate of -270% (Fig. 4h)[49]. The electromechanical durability can be further optimized by tuning the fiber's diameter. For example, a PUAL fiber of 1 mm achieves a resistance change rate of only -110% after sustaining 1000 stretching cycles with 60% strain at 80 °C, demonstrating the electromechanical versatility and significant application potential under thermal and mechanical stimuli (Supplementary Fig. 25).

## Static and dynamic electrical property enhancement of PUAL fiber at low temperatures

The PUAL fiber also shows autonomous electrical enhancement at low temperatures, due to the electrical connection variation as the TPU matrix volume changes (Fig. 5a). To verify the mechanism, PUA and PUAL fibers were frozen from 25 °C to −30 °C and naturally warmed to 25 °C for five cycles with their real-time conductivity recorded (Fig. 5b and Supplementary Fig. 26). The conductivity increased rapidly once the fiber started to be frozen until −30 °C. As the temperature naturally returns to 25 °C, the conductivity decreased to a value close to the initial conductivity, indicating that low-temperature conditions can

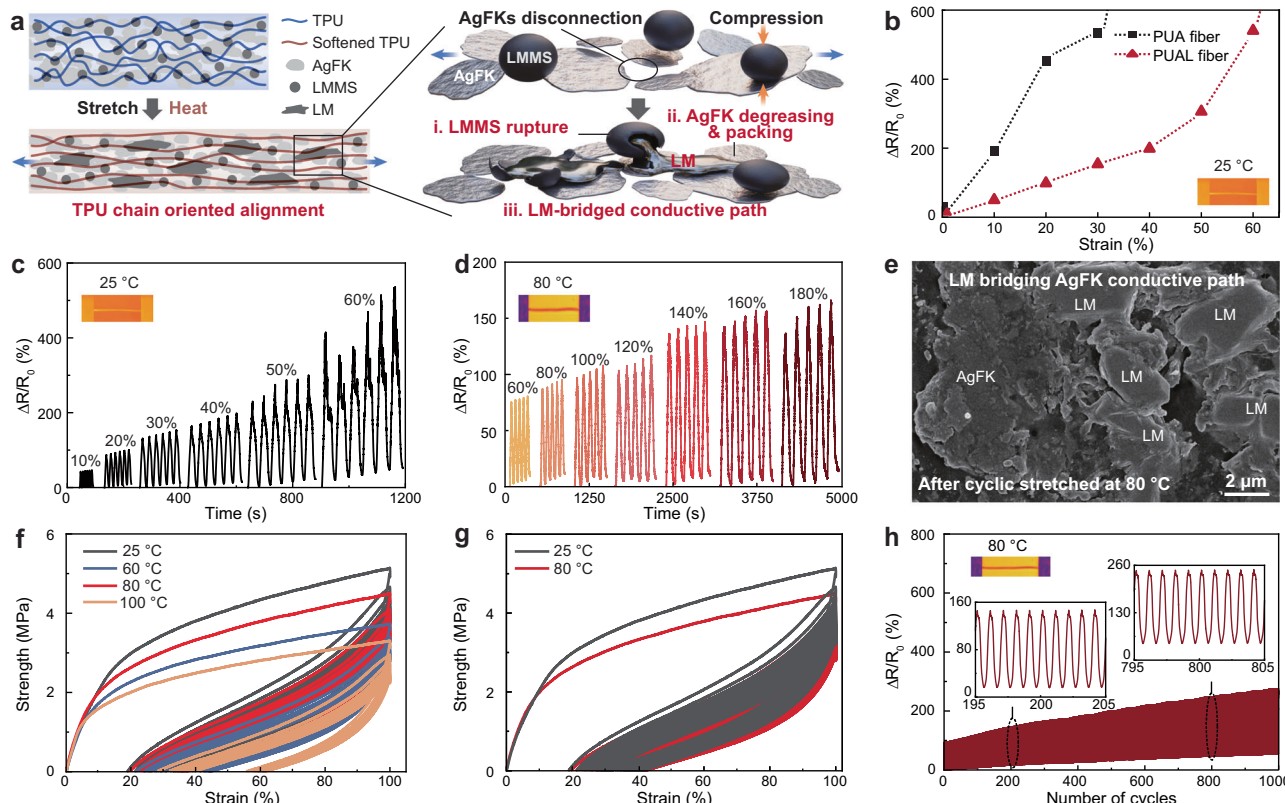

**Fig. 4 | Dynamic electromechanical stability enhancement of PUAL fiber at high temperatures. a** Schematic illustration of the thermal-stretching activation mechanism of the PUAL fiber by cyclic stretching under high temperatures. **b** Resistance change rates of PUA and PUAL fibers within 60% strain at 25 °C. **c, d** Resistance change rates of the PUAL fiber under (**c**) cyclic stretching within 60% strain at 25 °C, and (**d**) cyclic stretching within 180% strain at 80 °C, where 20 cycles were tested for each condition and the last 5–6 cycles were applied for clear plotting. **e** SEM image of the PUAL fiber revealing LMMSs' rupture to bridge AgFKs conductive paths under mechanical stretching and thermal stimulation. The main AgFK and released LM from ruptured LMMSs have been labelled. **f** Tensile loading-unloading curves of the PUAL fiber with 100% strain for 10 cycles, at 25 °C, and after heating at 60 °C, 80 °C and 100 °C for 30 min. **g** Cyclic loading-unloading tensile curves of the PUAL fiber with 100% strain for 100 cycles, at 25 °C and after heating at 80 °C for 30 min. **h** Electrical stability of the PUAL fiber over 1000 stretching cycles under 60% strain at 80 °C.

enhance the fiber's conductivity. PUAL fiber exhibits the optimal initial (~1070 S cm⁻¹ at 25 °C) and terminal conductivity (1160 S cm⁻¹ at −30 °C) in the first cycle, which was further improved to 1220 S cm⁻¹ at −30 °C in the fifth freezing cycle (Fig. 5b). Similar freezing-induced conductivity change behavior was observed in the PUA fiber (Supplementary Fig. 26) and other ECFs consisting of thermoplastic elastomers such as SEBS and SIS (Supplementary Fig. 27). In contrast, the electrical conductivity of a cast film with the same recipe did not exhibit conductivity enhancement trends when experiencing the same temperature variation due to the unstable conductive path (Supplementary Fig. 28). The results suggest a universal freezing-induced conductivity enhancement mechanism of elastic conductive fibers.

To further analyze the fiber's electrical enhancement mechanism at low temperatures, the cross-sectional area of the PUAL fiber when cooled down from 25 °C to −30 °C and then naturally warmed up to 25 °C was monitored using an in situ SEM. Slightly contracted cross-sections can be observed on the fiber at cryogenic conditions, which is reversed as the temperature rises back to 25 °C (Fig. 5c and Supplementary Fig. 29). Figure 5d and Supplementary Table 3 present the reversible change of the fiber's cross-sectional area from ~9.72 × 10⁴ μm² (25 °C) to ~9.58 × 10⁴ μm² (−30 °C), recovering to ~9.71 × 10⁴ μm² (25 °C). This contraction behavior is more intuitive in the axial length change of the fiber in TMA tests. As the PUAL fiber was repetitively frozen from 25 °C to −30 °C and then rewarmed to 25 °C for five cycles, a length contraction with a d$L/L_0$ of −0.18% was observed for the first cycle, and the deformation was largely reversible during the cyclic refrigeration, reaching a variation of d$L/L_0$ of −0.17%

at the 5th cycle (Fig. 5e). This demonstrates the reversible contraction of the fiber caused by the thermoplastic TPU matrix, which could continually promote the compact arrangement and electrical connections for both conductive fillers, increasing the fiber's conductivity, as verified by the gradually increased conductivity of the fiber at 25 °C and −30 °C during the five cycles of freezing-rewarming (Fig. 5b).

The effect of stretching stimulation under low temperatures on the fiber's electrical stability was also studied (Fig. 5a). To reveal dynamic electromechanical property variation, a natural warming process from −20 °C to 0 °C was applied to the frozen fiber samples, and the resistance variation of PUA and PUAL fibers was monitored under cyclic stretching with increasing strains (up to 80% strain, 20 cycles for each strain, with the last 10 cycles for clear plotting). The PUA fiber can sustain only 40% tensile strain from −20 °C to −5 °C, with a significant resistance variation of over 600% from −5 °C to 0 °C (Supplementary Fig. 30). In comparison, the PUAL fiber can sustain a higher tensile strain of 70% from −20 °C to 0 °C, and maintains much lower $\Delta R/R_0$ below 500% (Fig. 5f). This superior deformation tolerance and higher electrical stability of PUAL fiber well evidences the important roles of the LMMSs in repairing the fiber's electrical connection during the cold-stretching activation process, through deformation or rupture to release LM under the squeezing from both stretching and cold-induced TPU matrix contraction (Fig. 5a and Supplementary Fig. 31)[50]. This LMMS-activation-enabled electrical stability enhancement mechanism can also be supported by the obvious resistance change ($\Delta R/R_0$ of ~400%) increases in the first 10 stretching cycles at 70% strain, and then a significant resistance change reduction ($\Delta R/R_0$

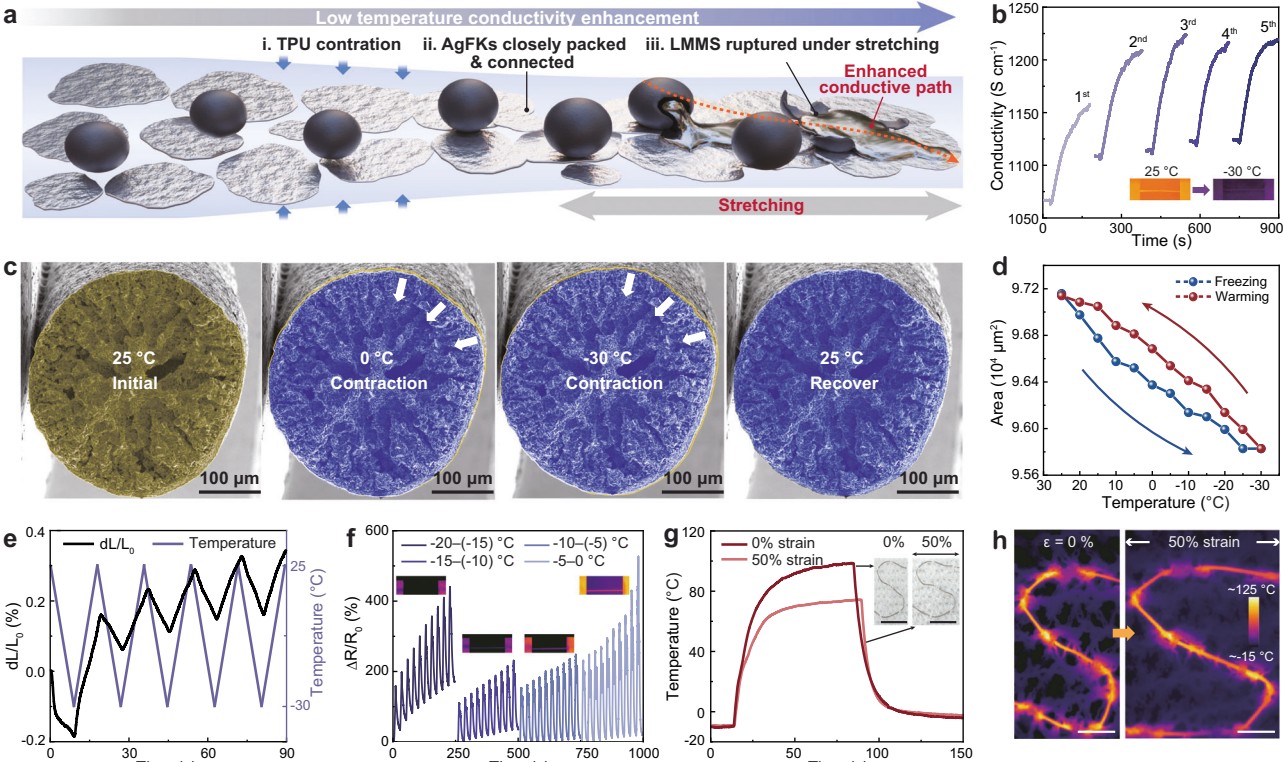

**Fig. 5 | Static and dynamic electrical property enhancement of PUAL fiber at low temperatures. a** Schematic diagram of the electrical enhancement mechanism activated by static or dynamic freezing. **b** Conductivity variation of the fiber in five freezing cycles from 25 °C to −30 °C (the start and end points of each curve correspond to the conductivity at 25 °C and −30 °C, respectively, with a two-hour interval between each cycle). **c** In situ SEM images of the fiber's cross-section at 25 °C, 0 °C, −30 °C, and back to 25 °C, revealing the low temperature-induced contraction behavior. The cross-sectional area in the initial state (25 °C) is marked in yellow, and the cross-sectional area during the freezing-natural warming process (0 °C, −30 °C, and 25 °C) is marked in blue. **d** Cross-sectional area changes of the fiber during the freezing and re-warming processes. **e** Cyclic TMA tests revealing the multiple deformation behavior of the fiber between 25 °C to −30 °C. **f** Resistance change rates of the fiber during cyclic stretching with 70% strain from −20 °C to 0 °C. **g, h** Electro-heating performance (**g**) and infrared thermogram (**h**) of the fiber woven into knitwear, functioning at −10 °C, under 0% and 50% strain. The inset of (**g**) shows the digital photo of the heater (scale bar: 5 cm). Scale bar in (**h**): 2 cm.

of ~200%) occurs in the next group of cycles. However, as the temperature increases and more stretching cycles are applied, both the volume expansion and elasticity degradation of the fiber would weaken the fillers' arrangement, resulting in reduced electrical stability. The same cold-stretching activation effect is also effective in the PUAL$_{20}$ fiber (see details in Supplementary Fig. 21), indicating a reliable dynamic electromechanical stability enhancement mechanism at low temperatures.

To sum up, the conductivity enhancement mechanism of the PUAL fiber at low temperatures is mainly due to the reversible TPU contraction-induced compact arrangement of conductive fillers, which promotes the electrical connection in the fiber. The electrical stability enhancement during the cyclic stretching activation process at low temperatures is because of the contraction/stretching-induced rupture of LMMSs providing additional electrical compensation[51] (Supplementary Fig. 13). This cold-stretching activation process shows a minimum effect on the fiber's mechanical properties (Supplementary Fig. 32), which maintains an ultimate strength of ~8 MPa and a fracture strain of 400% after being treated at ~20 °C for 30 min, and possesses slightly increased hysteresis after stretched with 100% strain for 100 cycles. Accordingly, the fiber can be woven into daily cotton textiles as electrical heaters to function via the joule heating effect even under cryogenic conditions (−30 to −10 °C) (Fig. 5g, h and Supplementary Fig. 33), which can be driven by 3 V voltage and sustain 50% tensile strain, presenting a pathway to realize thermal management in extreme conditions.

## PUAL fiber for wearable applications

To study the application scopes of PUAL fibers, the long-term durability in different environments (40% humidity, 80% humidity, and ultraviolet (UV) exposure) over 15 days was evaluated. The fiber can withstand environments with different humidity levels (conductivity maintained at around 1075 S cm$^{-1}$) and sustain long-term UV exposure (a slight increase in conductivity from 1072 S cm$^{-1}$ to ~1109 S cm$^{-1}$), as shown in Supplementary Fig. 34. This is attributed to the hydrophobicity and thermal stability of the TPU matrix that can protect the conductive fillers from oxidation or leakage. The leakage concern of the LM can be alleviated by observing the fiber's morphology and understanding the wet-spinning mechanism based on double diffusion and phase separation in the wet spinning, where the LMMSs are primarily encapsulated in the fiber, which mainly rupture inside the fiber upon mechanical or thermal stimulations (Supplementary Fig. 35). The leakage risk can be eliminated by additional encapsulation, such as using PDMS (Supplementary Fig. 36), which does not sacrifice the fiber's electromechanical advantages under heating/freezing-treatments and cold/thermal-stretching-activation (Supplementary Fig. 37).

Knitting or weaving capability is essential for electronic fibers in practical applications. The PUAL fiber can be knitted/woven into 2D and 3D structures, such as a snowflake and a hedgehog (Fig. 6a). The 3D-knitted hedgehog is formed with PUAL fibers as the body and nose, and fine cotton yarns as the face, where the body and nose are connected inside, showing the high robustness of the PUAL fiber to bear the weaving process and the high local stress. The 3D hedgehog can serve as a reliable conductive path for lighting up a light-emitting

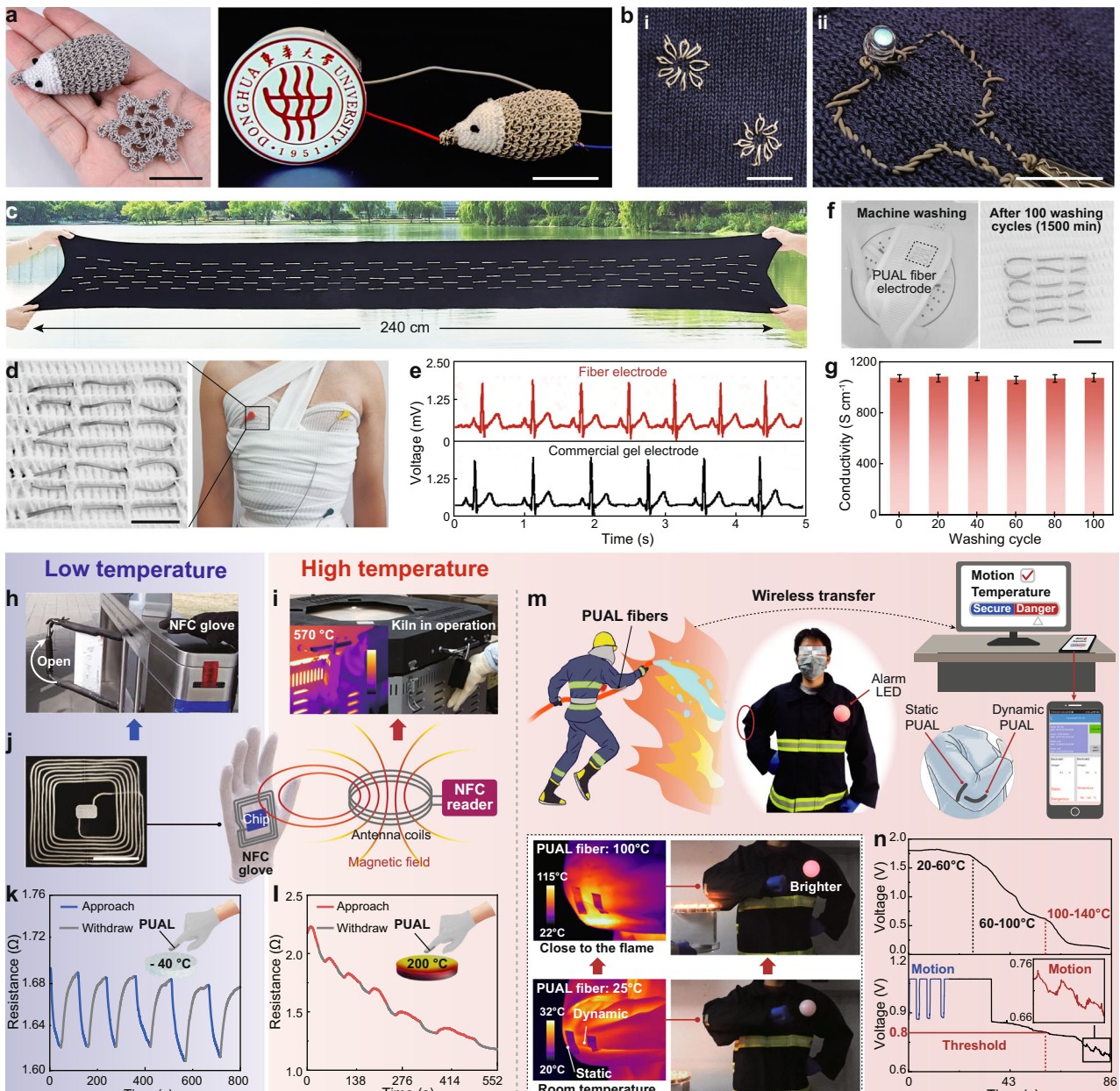

**Fig. 6 | Knittable ECFs for wearable applications. a** Hand-knitted hedgehog and snowflake with PUAL fibers, with the hedgehog serving as an electrode for lighting up an LED signage. **b** PUAL fibers stitched into a fabric as conductive wirings to light an LED. **c** A 1.6 m long fabric sewn with PUAL fibers sustaining 150% tensile strain. **d** PUAL fibers woven into medical bandages as ECG electrodes. **e** ECG signals measured by PUAL fiber electrodes and commercial solid conductive gel electrodes, respectively. **f** Digital photos showing machine-washing of the PUAL fiber electrode woven into medical bandages. **g** Conductivity of the PUAL fiber during the machine-washing process (five parallel samples were measured, and error bars represent the standard deviation). **h–j** A PUAL fiber NFC antenna for (**h**) outdoor access control operation in winter and (**i**) door access of a kiln in operation, with (**j**) digital photo of the PUAL NFC antenna and schematic showing its working mechanism. **k, l** The PUAL fiber woven into a glove for temperature indicating when approaching (**k**) cold and (**l**) hot objects. **m** PUAL fibers acting as thermomechanical-responsive electrodes in a high-temperature perceptive firefighter suit. **n** Voltage variation of the static (upper graph) and dynamic (lower graph) PUAL fibers in the firefighter suit. Scale bar: 2 cm, unless indicated in the figure.

diode (LED) signage (Fig. 6a). The PUAL fiber with mechanical and electrical stability can also be stitched into fabrics to form complex electrode patterns and light up an LED (Fig. 6b). Besides, this fiber can be sewn into a large elastic fabric (30 cm by 160 cm) and sustain 150% tensile strain (Fig. 6c), indicating its adaptability for large-scale processing and application. As an alternative to commercial hydrogel electrodes, the fiber can be woven into medical bandages for continuous electrocardiogram (ECG) monitoring, achieving comparable accuracy with commercial ECG electrodes (Fig. 6d, e and Supplementary Movie 1) and machine washability. This fiber maintains stable

conductivity (~1070 S cm⁻¹) even after being machine washed (GB/T 12490-2014) for 100 cycles (1500 min in total) with commercial detergent (Fig. 6f, g).

The PUAL fiber that goes through continuous heating-freezing or freezing-heating treatments shows the same conductivity change trends as those with single heating or freezing treatments (Supplementary Fig. 38). Thus, the effects of heating and freezing processes do not interfere with each other, enabling the PUAL fibers to be employed across temperatures to extend their application. Owing to the textile-integrating capability, elasticity and temperature-adaptive

conductivity of the PUAL fiber, we demonstrated the fiber's potential to convert daily textiles into various electronic textiles. First, the fiber can be processed as a near-field communication (NFC) antenna on protective gloves (Fig. 6h–j), which can complete NFC-controlled door access in cold weather (−5 °C) even after freezing at −30 °C for 5 h (Fig. 6h and Supplementary Movie 2), and can open the door of an electrical kiln in operation with an internal temperature around 570 °C (Fig. 6i and Supplementary Movie 2), demonstrating the application potential to adapt to scenarios with low or high temperatures. Second, a glove sewn with the PUAL fiber can indicate cold and warm objects via non-contact modality by differentiating the fiber's resistance variation, which could endow soft robots with temperature perceptivity to improve the safety of human-machine interaction (Fig. 6k, l). These demonstrations present wearable intelligent platforms built with PUAL fibers for non-contact operation, temperature indication, and communication, substantially improving ease-of-living comfort[52,53].

In addition, a wireless high-temperature warning firefighter suit is realized by integrating two PUAL fibers on the arm (Fig. 6m, Supplementary Note 5, Supplementary Table 4 and Supplementary Movie 3), where a tailored circuit was designed to quantify the fiber's resistance by measuring each fiber's partial voltage (Supplementary Fig. 39). One of the fiber parallel to the arm is the dynamic fiber, which can be stretched when the arm bends, to sense both the human motion and the environmental temperature qualitatively. Another fiber perpendicular to the arm is the static fiber that can perceive the environment temperature quantitatively as there is no interference from deformation. As shown in Fig. 6m, n, when the environmental temperature exceeds the threshold (100 °C, where the partial voltage of the dynamic fiber is 0.8 V, and the partial voltage of the static fiber is 0.6 V), an alarming LED on the firefighting suit becomes brighter. The temperature range and safety status of the firefighter can be transferred through Bluetooth to the mobile phone APP for autonomous communication. This demonstrates the application potential of PUAL fiber in textile-based wireless temperature-warning platforms to improve the safety of high-temperature operations.

## Discussion

Through wet-spinning-induced shear flow, we demonstrated a TPU-based elastic conductive fiber with highly oriented conductive fillers (AgFKs and LMMSs), which exhibits autonomously enhanced conductivity and electromechanical stability at low/high temperatures from −30 °C to 180 °C. We reveal that the oriented AgFKs dominate the formation of a stable conductive path, and the LMMSs improve the AgFKs' electrical connection. Heating can promote AgFK exposure, and freezing can contract the TPU matrix, promoting the AgFKs' arrangement and electrical connection to enhance the fiber's conductivity. This temperature-induced conductivity enhancement can be mechanically activated to release the liquid metal and improve the fiber's electrical stability under cyclic stretching, enabling a mechano-thermo-electrically cooperative mechanism to achieve a fiber with temperature-adaptive electromechanical stability and durability. The fibers can be produced on a large scale and stitched or knitted into various e-textiles, demonstrating wearable applications in biomedical electrodes, temperature indicators, temperature-adaptive electric heaters, NFC gloves, and intelligent firefighter suits. This work provides a universal mechano-thermo-induced electrical property enhancement mechanism for organic-inorganic composite elastic conductive fibers. It could inspire a common strategy for developing high-performance conductive fibers, broadening the application of e-textiles in wider scenarios.

## Methods
### Fabrication of PUL, PUA, and PUAL fibers
EGaIn (Ga, 75.5%, and In, 24.5% by weight; melting point -16 °C) was purchased from Dongguan Houjie Dingtai Metal Materials Co. Ltd.

AgFKs were obtained from Hebei Hangbei Metal Materials Co. Ltd. Thermoplastic polyurethane (TPU, $M_w$ = 130,000) were purchased from Dongguan Jinheng Plastic Co. Ltd. SIS, SEBS, N, N-dimethylformamide (DMF, ≥99.5%), and toluene (≥99.5%) were acquired from Shanghai Aladdin Biochemical Technology Co. Ltd. All chemicals and reagents are analytical grade without further treatment before usage.

TPU solution (18 wt%) was fabricated by adding TPU particles ($M_w$ = 130,000) into DMF solvent under mechanical stirring at 60 °C for 2 h. Then, at room temperature, LMMSs (the median particle size is 40 μm) were added to the TPU solution (where the mass ratio of LMMSs to TPU was 7) until evenly dispersed, to prepare the PUL fiber spinning solution. The PUA fiber spinning solution was prepared by adding AgFKs to the TPU solution until evenly dispersed, where the mass ratios of AgFKs to TPU were 2.5, 3, 3.5, and 4, respectively. After standing to defoam, the spinning solution was injected via a needle (inner diameter of 510 μm) in the coagulation bath of water, in which the flow rate was regulated by syringe pumps at 5–60 mL h$^{-1}$, and a winding collector was employed to collect the fibers. Building on the typical PUA spinning solution, LMMSs of different sizes (20 μm and 40 μm) were incorporated to fabricate the PUAL fibers (LMMSs' median particle size is 40 μm) and PUAL$_{20}$ (LMMSs' median particle size is 20 μm) through wet-spinning, following the same approach and parameters of the PUA fiber, where the mass ratios of LMMSs to AgFKs were 1, 1.5, 2, and 2.5, and the corresponding LMMS contents are 44 wt%, 54 wt%, 60 wt%, and 66 wt%, respectively.

### Electrical measurements
At room temperature, cyclic stretching of the fibers was carried out by a flexible electronic tester (FT 2000, Suzhou Shengte Intelligent Technology Co., Ltd., China) with a drawing speed of 30 mm min$^{-1}$ to acquire the real-time resistance using an electrometer (6510, Keithley, USA). To create a high and low temperature, the fiber with an effective length of 2 cm was placed on a hot plate or a polytetrafluoroethylene (PTFE) substrate pre-frozen with liquid nitrogen, where the fiber's temperature was recorded by an infrared camera. When reaching the specific temperature, a stretching stage controlled by a stepping motor was employed to stretch the non-fixed end of the fiber periodically, recording the fiber resistance simultaneously. The stretching speed was controlled by the following equation.

$$\pi r^2 L = V \tag{1}$$

$$L = vt \tag{2}$$

where $r$ refers to the syringe's diameter, $L$ refers to the lateral stretching distance, $V$ refers to the stroke volume, $v$ refers to the transverse stretching speed, and $t$ refers to the stretching time.

### Characterization
FTIR spectra of AgFKs were recorded using an FTIR spectrometer (Nicolet 6700, Thermo Fisher Scientific, USA). The morphology of the fiber, film, and filler samples was characterized using a field emission SEM (FE-SEM, SU 8010, Hitachi, Japan). The size evolution of the fiber was monitored through an in situ SEM (Thermo Scientific Scios2, Thermo Fisher Scientific, USA). TGA was carried out in an N$_2$ atmosphere using TG Instruments (TGA8000, PerkinElmer, USA) with a heating rate of 10 °C min$^{-1}$. X-ray diffraction was carried out by an X-ray diffractometer (Bruker D8 ADVANCE, Bruker, Germany). The melting enthalpy of the fiber was characterized with a differential scanning calorimeter (DSC8500, PerkinElmer, USA), with a heating rate of 10 °C min$^{-1}$. TMA was performed on a thermo-mechanical analyzer (402F3, NETZSCH, Germany) with a heating rate of 5 °C min$^{-1}$. DMA was performed on a dynamic thermo-mechanical analyzer (DMA1, Mettler

Toledo, Switzerland) with a heating rate of 5 °C min$^{-1}$, frequency of 1 Hz, and displacement of 50 μm. All thermal images were taken by a thermal imager (348 L, Fortric, USA). The variation of modulus was measured by the temperature field scanning mode of a rotary rheometer (MCR302e, Anton Paar GmbH, Austria) (heating rate of 10 °C min$^{-1}$, frequency of 1 Hz, strain of 1%). All the reported values of mechanical properties were the average based on at least three independent measurements for each sample. Accelerated washing tests were performed in a washing machine (XPB08-45-C, Shiyi Electric Appliance Co., Ltd., China) with 99.5 wt% water and 0.5 wt% commercial liquid detergent. The duration for each wash was 15 min, and the stirring speed was 500 rpm. The NFC chip and circuit were designed by Shanghai Feiju Microelectronics. An ECG recorder (PC-180B, Heal Force, China) was employed to record the ECG signals of a human subject. Informed consents were obtained from the subjects in both the ECG measurement and the firefighting suit experiment.

## Data availability
The data that support the findings of this study are available within this Article and its Supplementary Information. All data are available from the corresponding author upon request. All source data generated in this study have been deposited in Figshare (https://doi.org/10.6084/m9.figshare.28827701).

## Code availability
The custom code for high-temperature warning firefighter suit systems is available from the corresponding author upon request.

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

## Acknowledgements

The authors appreciate the funding support from the National Natural Science Foundation of China (52273244, 52103254, J.X.), Shanghai Sailing Program (24YF2700800, X.Z.), Fundamental Research Funds for the Central Universities (2232023Y-01, J.X.), and the China Postdoctoral Science Foundation (2023M730547, Yufan Z.).

## Author contributions

J.X. conceived the idea and supervised the project. Yue Z. designed/performed the experiments and collected the data. Yue Z., X.Z. and J.X. analyzed the data and discussed the results. Z.M., Z.Z., X.W., J.H., Yufan Z., W.L., L.Z., S.W., M.W. and Z.L. assisted in the experiment and result analysis. Yue Z., X.Z. and J.X. drew the figures and wrote the manuscript.

## Competing interests

The authors declare no competing interests.
