## [Transparent Peer Review file · Nature Communications]

A temperature-adaptive component-dynamic-coordinated strategy for high-performance elastic conductive fibers

Corresponding Author: Professor Jiaqing Xiong

Version 0:

Reviewer comments:

Reviewer #1

(Remarks to the Author)

I recommend major revision of this work.

- This manuscript highlights the use of Ag flakes (AgFKs) instead of Ag nanowires. Although AgFKs have been adopted for their ductility and ability to construct overlapping conductive pathways under shear force, Ag nanowires theoretically offer better stretchability. The authors should explicitly discuss whether Ag nanowires were tested and ruled out due to specific drawbacks, such as weaker alignment under wet-spinning conditions or lower conductivity under stress. Providing a quantitative comparison of the conductivity, stretchability, and scalability of Ag flakes and Ag nanowires will strengthen the justification for material selection and enhance the novelty of this study.
- The authors described shear-induced alignment during the wet-spinning process, but did not provide detailed evidence or metrics for this alignment. The authors should discuss whether any methods, such as adjustments in the flow rate or coagulation bath viscosity, were employed to control or enhance the alignment. The authors should also evaluate the degree of alignment using techniques such as SEM or XRD, and include metrics such as the orientation index or alignment factor to substantiate the claim. Visual evidence (e.g., SEM images) to demonstrate the differences between the aligned and non-aligned AgFK configurations should also be provided.
- Leakage of LMMSs under mechanical stress (e.g., rubbing or repetitive stretching) is a potential concern. The authors should address whether the LMMS rupture is confined to internal conductive pathways or if there is any external leakage. This could be supported by SEM images of the fiber post-deformation. Additionally, the manuscript should discuss strategies such as encapsulation or surface coating to mitigate the risk of leakage and ensure durability.
- The authors suggested that high temperatures induce lubricant removal and LMMS rupture, enhancing conductivity. However, is this process reversible upon cooling? If lubricant removal is not permanent, periodic high-temperature treatments may be required to maintain the performance.
- The softening behavior of the TPU matrix at 142°C warrants further investigation. The authors should relate this behavior to the TPU molecular structure using TGA or DSC data. The authors should also discuss how the softening temperature correlates with the mechanical stability and deformation of the fiber, particularly during high-temperature applications.
- The claim that “too few or too many LMMSs will sacrifice conductivity due to insufficient electrical bridging or distribution disturbance” needs further substantiation. The authors should provide plots of conductivity as a function of the LMMS concentration. The authors should also consider evidence such as SEM images to demonstrate how changes in LMMS concentration affect the microstructure and distribution of the conductive pathways.
- This manuscript attributes conductivity enhancement to lubricant removal, LMMS rupture, and AgFK alignment. To clarify their relative contributions, the authors should provide experimental evidence isolating each mechanism, such as tests with and without LMMSs or heating with unaligned AgFKs, and quantify the extent to which each mechanism contributes to the overall enhancement under different conditions (e.g., high vs. low temperatures).
- The manuscript suggests that low-temperature contraction enhances conductivity owing to AgFK alignment and LMMS

rupture, yet stretching does not produce the same effect. The authors should address the difference in filler behavior between these conditions and discuss whether the localized strain or uneven distribution explains the discrepancy.

- The resistance-strain relationship at 60°C (Figure 3f) shows a decrease in the resistance with increasing strain. This could be attributed to the enhanced filler alignment due to thermal softening. The authors validated this explanation using SEM images before and after the application of strain.
- The manuscript reports different strain values in Supplementary Figure 24 (60% and 80%) and the main text (70% strain sustainability from -20°C to 0°C). The authors should reconcile this discrepancy by clarifying whether the testing conditions differed, such as strain rate or temperature control.
- The proposed application of the fiber in high-temperature warning firefighter suits is innovative, but requires further elaboration. Please define the critical temperature threshold for determining the danger and describe the methodology used to establish it. Experimental data demonstrating how the fiber enhances firefighters' ability to perceive temperature changes compared with the baseline should also be provided.

Reviewer #2

(Remarks to the Author)

Reviewer #3

(Remarks to the Author)

In this manuscript, the authors introduce a strategy for high-performance elastic conductive fibers (ECF) adaptable to extreme temperatures. The ECF, composed of thermoplastic polyurethane (TPU), silver flakes (AgFKs), and liquid metal microspheres (LMMSs), was fabricated using a wet-spinning technique. Initially, these materials are randomly distributed within the solution; however, during the wet-spinning process, shear-induced alignment leads to the formation of regularly arranged conductive pathways. As a result, the ECF demonstrates significant mechanical and electrical performance enhancement. Furthermore, the materials used exhibit stability across a broad temperature range, enabling the presented ECF to perform reliably in extreme environments from -30°C to 180°C, showcasing outstanding functionality. However, as a reviewer, I find several areas in the manuscript that fall short of the standards required for publication in *Nature Communications*. Thus, I will reconsider its suitability after a major revision.

Major:

1. Lines 86–87 state, “The fiber’s conductivity can be significantly enhanced by improving the mass ratio of AgFKs to TPU, and not sacrifice the mechanical properties obviously (Supplementary Fig. 2).” Although the authors claim there is “obviously” no sacrifice to mechanical properties, the figure suggests an “obvious” compromise. This should be corrected, as even a seemingly minor exaggeration raises questions about the manuscript’s overall credibility.
2. The data in Fig. 2d and 2f seem insufficient. The authors demonstrate SEM images reflecting changes due to temperature; however, this does not seem adequately substantiated. Can temperature-dependent environmental SEM be captured in a single location? If feasible, additional similar images under consistent conditions would be helpful. Furthermore, although LM can be discerned in SEM images, I believe the inclusion of EDS data would clarify the findings.
3. In Fig. 3i, the authors claim that there is no “obvious resistance increase,” yet an “obvious” increase is indeed apparent. Moreover, the authors cite Fig. 3i as evidence of long-term usability. However, on closer inspection, a substantial notch appears at the top of each cycle, implying that the linearity of resistance change with strain could not be consistently maintained. This would preclude accurate strain estimation based on resistance, raising the question of how long-term usability is achievable. It seems unlikely; how do the authors plan to resolve this issue? If unresolvable, the claim of long-term usability seems unsustainable.
4. From my understanding of Fig. 5j and 5k, after one cycle of increased conductivity due to temperature change, the base resistance value itself appears altered. Is this application truly feasible, or is it suitable only for a single use? Likewise, the application depicted in Fig. 5l has issues. The image lacks clarity, making it difficult to discern whether the light is on or off in the unstretched state at room temperature, and in the supplemental movie, the “secure state” disappears too quickly; I could only confirm this by viewing at 0.3x speed. Furthermore, the application has critical flaws:
 - 1) If the wearer remains still, there is no indication of whether they are in a secure or dangerous state.
 - 2) As mentioned, if resistance changes due to temperature cause the device to enter a dangerous state, even upon returning to a safe environment, the device would not differentiate states. Does this mean the user must reinstall the device after each use? While single-use may hold meaning for firefighters, the application lacks significant impact.

I appreciate the intent to explore applications for extreme conditions, but the proposed application feels contrived. I

recommend identifying alternative applications.

Minor:

1. Are there any issues with liquid metal leakage? If knitted or woven into clothing, leakage would be unacceptable.
2. The manuscript presents samples repeatedly exposed to high or low temperatures only. I am curious about the conductivity change if a sample previously exposed to high temperatures were to be exposed to low temperatures or vice versa.
3. In Supplementary Fig. 8e, f, should the text reference 180°C instead of 100°C?

Reviewer #4

(Remarks to the Author)

The paper “A temperature-adaptive component-dynamic-coordinated strategy for high-performance elastic conductive fibers” reported the development of high-performance elastic conductive fibers (ECFs) that exhibit a temperature-adaptive, component-dynamic-coordinated strategy for enhanced conductivity, elasticity, and stability under extreme temperature conditions. The authors address a significant challenge in the field of electronic textiles (e-textiles): achieving a balance between conductivity, elasticity, knittability, and temperature tolerance. The study proposes a tricomponent-dynamic-coordination mechanism using thermoplastic polyurethane (TPU), silver flakes (AgFKs), and liquid metal microspheres (LMMSs), which are aligned through wet-spinning. Therefore, I would recommend its publication in Nature Communications after the minor revision.

1. Mechanistic discussions could be strengthened: While the paper provides a plausible explanation for the conductivity enhancement mechanisms, some claims require further quantification or validation. For example: The authors hypothesize that softening of the TPU matrix and LMMS rupture contribute to conductivity enhancement. However, quantitative data on LMMS rupture (e.g., how much liquid metal is released) and its correlation with conductivity changes could strengthen the argument. And the role of AgFK alignment in dynamic conductivity enhancement could be explored in more detail with additional analysis (e.g., SAXS analysis) or modeling.
2. Temperature adaptivity data could be strengthened: In the paper, the authors claimed that the ECFs autonomously enhance conductivity at extreme temperature, it would significantly bolster the paper's credibility if the authors could provide additional data on the long-term stability of the fibers when subjected to continuous thermal cycling between -30°C and 180°C.
3. Long-term durability and environmental stability: The paper demonstrates the fibers' stability under 1000 stretching cycles at 80°C and cyclic heating/cooling. However, it does not provide data on their long-term durability under environmental conditions such as humidity, UV exposure, or extended thermal cycling.
4. Could the authors please provide experimental evidence ensuring that no leakage of liquid metal occurs, even at the temperatures where the LMMSs are reported to rupture? This information is crucial for establishing the safety and practicality of the fibers for applications in harsh environmental conditions.

Version 1:

Reviewer comments:

Reviewer #1

(Remarks to the Author)

The authors have done a great job addressing my previous questions!

Reviewer #2

(Remarks to the Author)

Reviewer #3

(Remarks to the Author)

Thank you for the detailed and well-organized revision. The authors have addressed the reviewers' comments appropriately, and the revisions—particularly the enhancement of experimental data and clarification of mechanisms—contribute to a clearer presentation of the work. These updates also serve to better emphasize the novelty and relevance of the study. The TPU-based elastic conductive fiber with highly oriented conductive fillers (AgFKs and LMMSs) exhibits a unique combination of mechanical flexibility and directional conductivity. This distinctive characteristic makes it a promising candidate for various wearable applications, including biomedical electrodes, temperature indicators, temperature-adaptive electric heaters, NFC-enabled gloves, and intelligent firefighter suits, where both durability and responsive functionality are critical.

Reviewer #4

(Remarks to the Author)

The amendments and responses submitted show an apt consideration of the reviewers' comments, enhancing the overall quality and clarity of the research. The authors have adequately addressed the proposed comments by using the latest supplementary experimental data. The revised manuscript has been improved significantly and can be accepted in Nature Communications.

Response Letter for Manuscript ID NCOMMS-24-70055

Dear Reviewers:

We are sincerely grateful for your meticulous review and constructive feedback on our manuscript. Your constructive comments have been instrumental in enhancing the depth and rigor of our study. Given the reviewers' valuable suggestions, we have undertaken extensive revisions to our manuscript. The main revisions include:

- 1. Enhancing experimental data and refining discussion:** We have expanded our experimental data to provide a more robust foundation for our findings, ensuring that the results are all comprehensive and reliable.
- 2. Enhancement of mechanistic explanations:** We have refined the component-dynamic-coordinated conductive mechanisms of the PUAL fiber to ensure precision and clarity, enhancing the reader's understanding of our research.
- 3. Application demonstration refinement and further exploration:** We have refined our demonstrations of applications for extreme conditions and explored them further, making the applications more scientifically relevant and representative.
- 4. Revision of discussion section:** We have reorganized the Results and Discussion section to improve clarity and conciseness while preserving all essential data. We simplified the discussion about the PUAL₂₀ fiber as it does not affect the main discussion and conclusions.

The edited content that addresses the reviewer's comments has been highlighted in yellow in the revised manuscript. We appreciate the opportunity to revise and resubmit our work for consideration in *Nature Communications*. We have presented a comprehensive point-by-point response to all comments provided by the reviewers. We believe that our manuscript is now better positioned to contribute to the scientific discourse in your esteemed journal. If need more information, please feel free to contact us.

Jiaqing Xiong

Professor, PhD

State Key Laboratory of Advanced Fiber Materials, College of Textiles, and Innovation Center for Textile Science and Technology, Donghua University
2999 North Renmin Road, Shanghai 201620, China

Email: jqxiong@dhu.edu.cn

Reviewers' Comments

Reviewer #1 (Remarks to the Author): I recommend major revision of this work.

Response: Thank you for your valuable comments and for offering the opportunity to revise our manuscript. We have added data and revisions to address the comments point by point as below.

Comment 1.1: This manuscript highlights the use of Ag flakes (AgFKs) instead of Ag nanowires. Although AgFKs have been adopted for their ductility and ability to construct overlapping conductive pathways under shear force, Ag nanowires theoretically offer better stretchability. The authors should explicitly discuss whether Ag nanowires were tested and ruled out due to specific drawbacks, such as weaker alignment under wet-spinning conditions or lower conductivity under stress. Providing a quantitative comparison of the conductivity, stretchability, and scalability of Ag flakes and Ag nanowires will strengthen the justification for material selection and enhance the novelty of this study.

Response: Thank you for your concerns regarding the reason for selecting Ag flakes (AgFKs) instead of Ag nanowires (AgNWs). The reasons include the following: First, equivalent quality AgNWs are much more expensive than AgFKs¹ (AgFK: \$1.4/g, AgNW: \$550/g), which goes against the idea of simple, efficient, and low-cost wet-spinning of elastic conductive fibers (ECFs). Second, AgNWs are usually hydrophilic and show poor compatibility with hydrophobic polymer matrixes (such as the thermoplastic polyurethane (TPU) used in this paper)². In general, hydrophobic pre-treatments are required for AgNWs before the blending, which increases the process complexity and sacrifices the conductivity. To compare the properties of AgFK and AgNW for this elastic conductive fiber application, the following experiments were supplemented. By replacing AgFKs using AgNWs with the same content, we directly blended commercial AgNWs with the TPU/LMMS system for wet-spinning, and compared its morphology and electrical performance with our AgFK-based fiber. As shown in **Fig. R1a-c**, the TPU/AgNWs/liquid metal microspheres (LMMS) composite fibers show obvious AgNWs aggregation due to the poor compatibility between AgNWs and our existing system of TPU/LMMS. Meanwhile, the AgNW-based fiber shows worse electrical performance compared with our AgFK-based fibers (**Fig. R1d**). The conductivity of AgNW-based fibers is initially too small to be detectable, and increases as heated to 180 °C ($\sim 3 \times 10^{-5}$ S cm⁻¹), due to the heating-induced conductivity enhancement mechanism. In contrast, the AgFK-based fiber shows an initial conductivity of ~ 1070 S cm⁻¹, which can increase to 3020 S cm⁻¹ when heated to 180 °C (**Fig. R1e**). Thus, we choose AgFKs instead of AgNWs in considering simplicity, high efficiency, and high performance for ECF fabrication. The related data and clarification have been supplemented in **Supplementary Figure 2** and the manuscript (**page 4**).

Fig. R1. SEM photos and conductivity variation of TPU/AgNW/LMMS and TPU/AgFK/LMMS (PUAL) fiber. (a) Surface, (b) cross-section, and (c) longitudinal-section SEM images. (d) Conductivity variation of TPU/AgNW/LMMS fiber from 150 °C to 180 °C, where the conductivity is too small to be detectable below 150 °C. (e) Conductivity variation of PUAL fiber from 25 °C to 180 °C.

Comment 1.2: The authors described shear-induced alignment during the wet-spinning process, but did not provide detailed evidence or metrics for this alignment. The authors should discuss whether any methods, such as adjustments in the flow rate or coagulation bath viscosity, were employed to control or enhance the alignment. The authors should also evaluate the degree of alignment using techniques such as SEM or, and include metrics such as the orientation index or alignment factor to substantiate the claim. Visual evidence (e.g., SEM images) to demonstrate the differences between the aligned and non-aligned AgFK configurations should also be provided.

Response: Thank you for your insightful comments on the alignment (orientation) degree of AgFKs dependent on the wet-spinning process. Carefully studying the AgFK-oriented behavior is important for

understanding the mechanism of filler orientation-enhanced electromechanical properties of the fiber. We have supplemented the evidence of the shear-induced AgFK orientation and methods to enhance the AgFKs' orientation. Since our spinning coagulation bath is water, there is no viscosity involved, so we focus on evaluating the effect of spinning speed on the AgFKs' orientation. The orientation degree can be evidenced by SEM or wide-angle x-ray scattering (WAXS). Specifically, we supplemented the following three sets of experiments and discussions.

(1) AgFKs orientation analysis by morphology analysis: To clarify the formation mechanism and quantify the AgFKs' orientation, we cut the TPU/AgFK (PUA) and PUAL fibers along the axial direction, and took SEM images of their longitudinal section to compare with those of cast films with the same compositions. As shown in **Fig. R2**, in the cast films, the AgFKs are randomly embedded in the TPU matrix without obvious orientation. In comparison, all the fiber samples show a higher orientation of AgFKs in both longitudinal-section and cross-section, indicating that the shear effect during the wet-spinning is crucial for improving the AgFKs orientation. To quantify the oriented degree, about 50 AgFKs were randomly taken from the longitudinal-section SEM images to measure the angle between their long axis direction and the fiber's axial direction, where the angle less than 45 degrees was counted as oriented (**Fig. R3a**). The ratio of oriented AgFKs is calculated and compared for the representative fiber samples (PUA and PUAL) (**Fig. R3b**). Since the film sample does not have an axis, it is not suitable for this calculation. It is observed that all the fiber samples achieved high AgFK orientations over 80%. By observing the morphology of fiber and cast film, it is evidenced that wet-spinning is crucial for improving the orientation of AgFKs. This orientation is important for increasing both the mechanical and electrical properties of the samples, as evidenced in **Fig. R3c, d** where higher ultimate strength and electrical conductivity were observed from the fibers instead of the films. The related data and clarification have been supplemented in **Supplementary Figures 8 and 9** and the manuscript (**page 6**).

Fig. R2. SEM images of PUA and PUAL fibers and cast films with the same compositions. (a) PUA fiber and cast film. (b) PUAL fiber and cast film. From left to right are the axial longitudinal section, cross-section, and in-plane cross-section of the samples. The main AgFKs in the samples are marked in bright yellow.

Fig. R3. Orientation of PUA and PUAL fiber and conductivity and mechanical properties of PUA and PUAL fibers and films. (a) Schematic diagram of the calculation method of AgFKs' orientation in fiber's longitudinal section. (b) Axial orientation of AgFKs within PUA and PUAL fibers, calculated from the analysis of SEM images. (c, d) Mechanical and electrical properties of (c) PUA and (d) PUAL fibers compared with the films with the same compositions.

(2) Effect of wet-spinning speed on AgFKs orientation of PUAL fibers:

To illustrate how shearing force and wet-spinning speeds affect the orientation of AgFKs along the fiber's axial direction, we conducted a hydrodynamic finite element analysis of AgFKs in a unit cell model, by multi-field coupling of laminar flow and solid mechanics, with transient solver and moving mesh to track the AgFK movement. The simulation results show that AgFK's orientation is closely related to the stresses induced by the shear effect at the syringe funnel (**Fig. R4a**). The stresses on AgFK parallel to the flow direction are significantly lower than those in the perpendicular direction (the cloud diagrams are processed in absolute value for ease of presentation), causing AgFKs in the perpendicular direction of the flow are prone to rotate, while those in the parallel direction of the flow are relatively stable (**Fig. R4b**). Therefore, AgFKs in the perpendicular flow direction tend to orient parallel to the flow direction. Such an orientation can be attributed to the stress difference acting on the AgFK, which is directly related to the applied inlet velocity which has the same physical meaning as the wet-spinning speed. The transformation formula for the two is as follows.

$$v_{inlet} = \frac{3600 v_{wet\ spinning}}{\pi r^2}$$

As shown in **Fig. R4c**, a systematic study of wet-spinning speed (from 5 mL h⁻¹ to 60 mL h⁻¹) in the initial state reveals obvious stress differences in vertical and parallel can be produced with a spinning speed from 20 mL h⁻¹ to 50 mL h⁻¹, which could be a driving force to improve the AgFKs orientation. The above simulation results aid in illustrating that the flow-induced shearing force in wet-spinning leads to the high orientation of AgFKs along the fiber's axial direction.

Experimentally studying the wet-spinning speed effect on AgFK orientation, we prepared PUAL fibers using four different sets of wet-spinning speeds (5 mL h⁻¹, 20 mL h⁻¹, 40 mL h⁻¹, and 60 mL h⁻¹). The cross-sectional morphology (**Fig. R5**) suggests that the flow rate affects both fiber uniformity and filler orientation. When the spinning speed is too slow (e.g., 5 mL h⁻¹), AgFKs have a relatively lower orientation since the small speed gradient (between the ink inside the wide syringe and the thin needle) renders a small shear rate, leading to irregular cross-sections of the fibers (**Fig. R5a**). However, a too-fast speed (e.g., 60 mL h⁻¹) will cause turbulent flow, which increases the stress difference between the edge and the center of the solution flow, disturbing the filler orientation and resulting in fibers with a non-uniform diameter (**Fig. R5d**)³. In comparison, suitable wet-spinning speeds (e.g., 20 mL h⁻¹ and 40 mL h⁻¹) form fibers with a circular cross-section, as well as high AgFK orientation inside the fillers thanks to moderate shear force (**Fig. R5b, c** and **Fig. R6a**). In addition, the fibers with uniform diameter demonstrate higher mechanical properties than that of the fiber obtained under too slow (e.g., 5 mL h⁻¹) and too fast (e.g., 60 mL h⁻¹) wet-spinning speeds (**Fig. R6b, c**), and the fibers with fast (e.g., 20–60 mL h⁻¹) wet-spinning speeds demonstrate higher electrical properties (**Fig. R6d**).

Therefore, a suitable spinning speed (20 mL h^{-1} to 40 mL h^{-1}) is essential to enhance the diameter uniformity, filler orientation and mechanical properties of the fibers, and 20 mL h^{-1} was chosen as the optimal wet-spinning speed to prepare the fiber samples in this work. The related data and clarification have been supplemented in **Supplementary Figures 11 and 12** and the manuscript (**page 7**).

Fig. R4. Hydrodynamic analysis and orientation calculation of AgFKs under different wet-spinning speeds. **(a)** Stress diagram of AgFKs in X direction (wet-spinning flow direction) at a flow rate of 20 mL h^{-1} in the initial state. **(b)** Flow-induced orienting AgFKs at a constant time scale at a flow rate of 20 mL h^{-1} . **(c)** Maximum AgFKs stress under different wet-spinning speeds in the initial state.

Fig. R5. SEM images of PUAL fibers prepared with different wet-spinning speeds. **(a)** 5 ml h⁻¹. **(b)** 20 ml h⁻¹. **(c)** 40 ml h⁻¹. **(d)** 60 ml h⁻¹. The AgFKs in the enlarged images have been marked in yellow.

Fig. R6. Orientation and properties of PUAL fibers prepared with different wet-spinning speeds. **(a)** The axial orientation of AgFKs calculated by analyzing longitudinal-section SEM photos of the fibers. **(b)** Diameter uniformity (five parallel samples were measured, and error bars represent the standard deviation of the mean). **(c)** Mechanical properties (five parallel samples were measured, and error bars represent the standard deviation of the mean). **(d)** Electrical conductivity (five parallel samples were measured, and error bars represent the standard deviation of the mean).

(3) Orientation factor of AgFKs calculated by WAXS: WAXS is a common technique to study the orientation of materials. To further confirm the AgFK's orientation, WAXS was performed for the PUAL fiber. First, three obvious diffraction peaks corresponding to the lattice plane of (111), (200) and (220) were observed in the WAXS profiles of PUAL fiber (**Fig. R7a**)⁴. Meanwhile, a strongly angle-dependent 2D WAXS diffraction pattern was demonstrated on the PUAL fiber (**Fig. R7b**)⁵, suggesting the existence of AgFKs' orientation. Furthermore, Hermann orientation factor can be calculated according to the following equations⁶.

$$f = \frac{3\langle \cos^2 \varphi \rangle - 1}{2}$$

$$\langle \cos^2 \varphi \rangle = \frac{\int_0^{2\pi} I(\varphi) \sin \varphi \cos^2 \varphi d\varphi}{\int_0^{2\pi} I(\varphi) \sin \varphi d\varphi}$$

The value of f is between 0 (for random orientation) and 1 (for a perfectly oriented sample). $I(\varphi)$ is the 1D intensity distribution along with the azimuthal angle φ of the (111) and (200) planes of AgFKs (**Fig.**

R7c, d). In this paper, the orientation factor f of PUAL fiber was calculated to be 0.25, indicating a certain degree of orientation existed in the fiber. The relatively low orientation factor is because AgFKs align well along the axial direction but have varied orientations along the radial direction. This anisotropic orientation cannot be well reflected in WAXS, so the WAXS result can be regarded as supplementary evidence, and the directly calculated orientation from the SEM images would be more intuitive and convincing to reflect the AgFKs' orientation level in fiber samples. The related data and clarification have been supplemented in **Supplementary Figure 10** and the manuscript (**page 7**).

Fig. R7. WAXS of PUAL fiber. (a) WAXS curve. (b) 2D WAXS pattern. (c) The azimuthal plot of scattering at (111) along the φ direction in the region of 0° to 360° . (d) The azimuthal plot of scattering at (200) along the φ direction in the region of 0° to 360° .

Comment 1.3: Leakage of LMMSs under mechanical stress (e.g., rubbing or repetitive stretching) is a potential concern. The authors should address whether the LMMS rupture is confined to internal conductive pathways or if there is any external leakage. This could be supported by SEM images of the fiber post-deformation. Additionally, the manuscript should discuss strategies such as encapsulation or surface coating to mitigate the risk of leakage and ensure durability.

Response: Thank you for your valuable comments. The concern of LMMS leakage can be allayed through understanding the fiber formation mechanism based on double diffusion and phase separation in the wet spinning⁷. Specifically, the solvent in the spinning solution diffuses into the coagulation bath, while the water (coagulation bath) diffuses into the fiber, inducing phase separation to occur in the TPU matrix, which triggers the TPU molecular chains entanglement to solidify into fibers. This process is accompanied by the elastomer matrix contraction, wrapping most of the conductive materials and with a small portion of exposure, as demonstrated by SEM images of the PUAL fiber surface at 25 °C (**Fig. R8**). This structure could prevent LMMS leakage and ensure extra electrical connection.

In this study, the leakage of LM mainly exists under the stimulation of certain mechanical pressure and high temperature as shown in **Fig. R8**, where LMMS ruptured to compensate the conductive paths of AgFKs in PUAL fiber. As observed in **Fig. R8**, under static heat stimulation or moderate mechanical stimulation, the LMMS rupture on the surface is negligible. Only under severe mechanical deformation, such as when repeatedly handling the PUAL fibers in the experiments, leakage of LM is observed on the fiber surface, however, it does not affect the fiber's practicability. We have demonstrated that the fiber can sustain 1000 stretching cycles with 60% strain at 80 °C with a resistance change ($\Delta R/R_0$) of only ~270% (**Figure 4h**). During the sewing and weaving, no observable LM is found to contaminate the cloth or tools, and the fiber maintains stable conductivity even after 1500 minutes of machine washing (**Fig. R9, Figures 6a, c, and f**).

Encapsulation would be an effective strategy to avoid the leakage of LMMSs under mechanical stress. For example, we encapsulated PUAL fibers using 50 wt% polydimethylsiloxane (PDMS) solution. The encapsulated fiber shows a smooth surface with a uniform PDMS layer of about 10 μm (**Fig. R10**). Meanwhile, PDMS has excellent electrical insulation and weather resistance, as well as hydrophobicity to protect the fiber from external erosion, which can improve the fiber's stability⁸. The encapsulation does not sacrifice the fiber's electrical properties. The similar conductivity-enhancement behavior as the original fiber was observed, which increases from 1070 S cm^{-1} (25 °C) to 1208 S cm^{-1} (-30 °C) and 1608 S cm^{-1} (180 °C) and 2463 S cm^{-1} (240 °C) under cooling and heating conditions (**Fig. R11a, b**), respectively. In addition, the encapsulated fiber can also be stretching-activated to improve its mechano-electrical and thermoelectrical stability, maintaining stable resistance ($\Delta R/R_0$ of ~240% at 60% strain over 1000 stretching cycles, 80 °C, and ~240% at 80% strain, ~ -15 °C) (**Fig. R11c, d**), with similar trends align with the value without encapsulation. Excellent interfacial stability was observed on the fiber's cross-section after 1000 cycles of stretching deformation (**Fig. R11e, f**), indicating this encapsulation is an acceptable strategy for improving the fiber's electromechanical stability.

It should be noted that the encapsulated fiber only allows electrical connection at its ends, which reduces the fiber's application convenience in a certain. Thus, the encapsulation can be considered in specific applications. The related data and clarification have been supplemented in **Supplementary Figures 35-37** and the manuscript (**page 16**).

Fig. R8. SEM images of PUAL fiber under different temperatures, with or without tensile strain, indicating that LMMSs mainly rupture upon mechanical or thermal stimulations. The main deformed & ruptured LMMSs under different temperatures without tensile strain have been marked by red circles, and the main ruptured LMMSs under different stretching conditions have been marked in bright yellow.

Fig. R9. Demonstration of no visible leakage of LMMSs from PUAL fiber during crocheting and sewing. (a) Hand-crochet of PUAL fiber. (b) A fabric sewn with PUAL fibers. (c) The PUAL fiber ECG electrode in medical bandages after washed for 1500 minutes. Scale bar: 2 cm.

Fig. R10. SEM images and EDS mapping of PUAL fiber encapsulated with PDMS. (a) Cross-sectional SEM and EDS mapping. (b) Enlarged cross-sectional SEM. (c) Surface SEM. The Ga and In signals at the top left of the fiber are from the previous contamination of the stub.

Fig. R11. Properties of PDMS-encapsulated PUAL fiber. (a) Conductivity variation from 25 °C to 240 °C. (b) Conductivity variation from 25 °C to -30 °C. (c) Resistance changes under cyclic stretching with 60% strain at 80 °C over 1000 stretching cycles. (d) Resistance changes under cyclic stretching with 80% strain from -20 °C to 0 °C. 20 cycles were tested for each temperature range, with the last 10 cycles for clear plotting. (e) Surface morphology of the encapsulated fiber after cyclic stretching with 60% strain at 80 °C over 1000 stretching cycles. (f) Cross-sectional SEM image of the encapsulated fiber after cyclic stretching with 60% strain at 80 °C over 1000 stretching cycles.

Comment 1.4: The authors suggested that high temperatures induce lubricant removal and LMMS rupture, enhancing conductivity. However, is this process reversible upon cooling? If lubricant removal is not permanent, periodic high-temperature treatments may be required to maintain the performance.

Response: Thank you for your valuable comments. We claim that both the lubricant removal of AgFKs (AgFK degreasing) and the LMMS rupture involved in the conductivity enhancement at high temperatures (static) are irreversible. We have conducted experiments to verify the permanency of both effects separately and as a whole, by long-term conductivity measurements. First, to distinguish the LMMS rupture and AgFK lubricant removal effect at high temperatures, we studied the temperature threshold of the LMMS rupture in promoting the fiber's conductivity. As shown in **Fig. R12**, when the temperature is below 150 °C, the LMMS rupture does not function in increasing the fiber's conductivity. It is verified by the deformation/rupture trend of the LMMSs as temperature rises (**Fig. R13a, Figure 3f and Supplementary Figure 15**), which does not recover after natural cooling. Therefore, we keep the temperature below 100 °C to study the AgFKs degreasing effect.

As reported by literature, the surface lubricants of AgFKs can be removed by heating, which is irreversible⁹. Herein, SEM images of AgFK of PUAL fiber (heated from 25 °C to 100 °C to 180 °C, and naturally cooling to 25 °C) indicated that rougher AgFK surface induced by the elevated temperature did not recover as the temperature drops (**Fig. R13b, Figure 3d and Supplementary Figure 15**). Cyclic heating (either continuous or across a large time frame) is conducted to understand the mechanism of lubricant removal. **Figure 3g (Fig. R14a)** shows the cyclic heating treatments (25 °C–100 °C–25 °C) of PUAL fiber, where continuous resistance changes of the fiber dependent on the temperature and heating number were observed. In the first cycle, the fiber conductivity ($\sim 1070 \text{ S cm}^{-1}$) initially shows a slight decrease and then sharply increases to 1361 S cm^{-1} , while in the last four cycles, the fiber conductivity always reduces from a further increased initial value (1651 S cm^{-1} – 1741 S cm^{-1}) to a relatively stable value of 1431 S cm^{-1} – 1470 S cm^{-1} . It suggests that AgFKs degreasing dominates the electrical enhancement in the first cycle, and most of the lubricants have been removed. Thereafter, when the fiber was cyclically cooled from 100 °C to 25 °C before the next heating cycle, the cooling-induced fiber's contraction enhanced the electrical connections between the conductive fillers, rendering an increased conductivity. During each heating cycle except the 1st cycle, the fiber's conductivity decreases due to the TPU's thermal expansion to reduce the fillers' electrical connections. However, comparing each ending point of the heating curves, which represents the conductivity at 100 °C, the matrix expansion effect can be controlled by fixing the temperature. The only factor for the increasing trend in the ending point conductivity (from 1361 S cm^{-1} to 1470 S cm^{-1}) is the gradual removal of the lubricants from the AgFKs.

To further verify the permanency of the cyclic lubricant removal, we extend the cyclic heating to a 7-day timeframe. As shown in **Fig. R14b**, PUAL fibers' conductivity also shows a large increase in the first cycle (from $\sim 1070 \text{ S cm}^{-1}$ to 1361 S cm^{-1}), then increases gradually in the following cycles, and eventually remains a stable value of $\sim 1470 \text{ S cm}^{-1}$ after the 7th cycle. This behavior aligns with the continuous heat cycling result, and the saturated conductivity further proves the permanent removal of AgFK's lubricant layer.

Although the heating-induced AgFK degreasing and LMMS rupture are irreversible, this statically heating-induced conductivity enhancement mechanism still provides a new strategy to achieve high-performance ECFs. Meanwhile, in most application scenarios, the fiber will not continuously be subjected to a very high-temperature condition. The PUAL fiber's conductivity variation can be continuously achieved for multiple heating cycles, which could allow its application for thermal perception (**Fig. R15, Figure 6l**). Specifically, we can take advantage of the multiple removal feature of AgFKs lubricant to realize non-contact perception of hot objects. The fiber sewn on a glove first shows a tendency to increase (TPU matrix expansion) and then decrease its resistance (massive removal of AgFKs lubricant and TPU matrix contraction) in each cycle upon approaching and moving away from a hot object. The related data and clarification have been supplemented and refined in the manuscript (**pages 6, 8 and 17**) and **Supplementary Figures 4 and 16 and 17**.

Fig. R12. Conductivity of PAL fiber from 150 °C to 180 °C.

Fig. R13. SEM images of LMMSs (a) and AgFK (b) of PUAL fibers at different temperatures (25 °C, 100 °C, 180 °C and natural cooling from 180 °C to 25 °C).

Fig. R14. Conductivity variation of the PUAL fiber. (a) Cyclic conductivity variation between 25 °C to 100 °C, with a two-hour interval between each cycle. The start and end points of each segment of the curve correspond to the conductivity at 25 °C and 100 °C, respectively. (b) The fiber's conductivity variation during multi-cyclic heating over a week from 25 °C to 100 °C once every day (five parallel samples were measured, and error bars represent the standard deviation of the mean).

Fig. R15. Cyclic resistance variation when a glove sewn with a PUAL fiber periodically approaches and moves away from an object of 200 °C.

Comment 1.5: The softening behavior of the TPU matrix at 142 °C warrants further investigation. The authors should relate this behavior to the TPU molecular structure using TGA or DSC data. The authors should also discuss how the softening temperature correlates with the mechanical stability and deformation of the fiber, particularly during high-temperature applications.

Response: Thank you for your insightful comment and suggestion. TPU is a linear polymer with a molecular chain consisting of alternating flexible soft chain segments and rigid hard chain segments. This structure allows microphase separation to occur between the soft and hard chain segments under suitable conditions¹⁰. To the best of our knowledge, the softening temperature of elastomers depends on the degree of microphase separation, and the softening temperature of TPUs without microphase separation is only about 70 °C, while it can reach up to 130–150 °C if there is microphase separation¹¹. **Figure 2b (Fig. R16a)** indicates that the TPU used in this work possesses a high softening temperature of 142 °C, ensuring the thermomechanical stability of the fiber and implying that there might be an existence of microphase separation, as verified by thermalgravimetric analysis (TGA), where two-step weightless behavior existed, and small-angle X-ray scattering (SAXS) results with a peak of scattering vector (q) (**Fig. R16b, c**)^{12, 13}, laying the foundation for our study.

In this paper, the effect of the TPU's softening temperature on the mechanical stability of the fiber can be briefly illustrated as follows. When the TPU matrix is in the glassy state (< -34 °C), the molecular chain segment motion is limited, thus, the fiber shows low deformability with certain stiffness and brittleness (**Fig. R16d**). As the temperature increases (-34 °C–142 °C, **Figures 2b, c**), the polymer reaches a rubbery state that allows the molecular chains to change their conformation by local segment

rotation under external stress, rendering the fiber high elasticity. If the temperature continues to increase to the softening temperature, the molecular backbone of TPU begins to slip, leading to irreversible fiber deformation. Our work focuses on the mechanical-electrical properties of the TPU-based composite fibers under the rubbery state (the temperature of $-34\text{ }^{\circ}\text{C}$ – $142\text{ }^{\circ}\text{C}$, **Fig. R16a, d**). For example, the PUAL fiber after cyclic stretching (60%–180% strain; 60%–70% strain) at different temperatures ($60\text{ }^{\circ}\text{C}$ – $100\text{ }^{\circ}\text{C}$; $-20\text{ }^{\circ}\text{C}$ – $0\text{ }^{\circ}\text{C}$) exhibited favorable electromechanical properties (**Figures 4d, e and Supplementary Figures 21, 22 and 30**). The main application of PUAL fibers (high-temperature warning for firefighting suits) is also located within the softening temperature ($142\text{ }^{\circ}\text{C}$) (**Figure 6m**). Therefore, the mechanical properties and stability analysis for PUAL fibers under high temperatures are convincing. The related data and clarification have been supplemented in **Supplementary Figures 4 and 21, 22 and 30** and the manuscript (**page 4**).

Fig. R16. Thermodynamic properties of TPU fiber. **(a)** Thermo-mechanical analysis (TMA) curves reveal the softening behavior. **(b)** SAXS curve. **(c)** TGA curve. **(d)** Temperature-dependent $\text{tan}\delta$ (ratio of loss modulus to storage modulus) in dynamic mechanical analysis (DMA).

Comment 1.6: The claim that “too few or too many LMMSs will sacrifice conductivity due to insufficient electrical bridging or distribution disturbance” needs further substantiation. The authors should provide plots of conductivity as a function of the LMMS concentration. The authors should also consider evidence such as SEM images to demonstrate how changes in LMMS concentration affect the microstructure and distribution of the conductive pathways.

Response: Thank you for your valuable suggestions. In the manuscript, we provided the conductivity data of PUAL fibers prepared with different ratios of LMMSs to AgFKs (**Fig. R17, Supplementary Figure 7**). Based on your suggestion, we converted the LMMSs : AgFKs ratio into the LMMS weight content and made additional conductivity plots, where the fiber with 60 wt% LMMS (LMMSs : AgFKs = 2) demonstrates the optimal conductivity of $\sim 1070 \text{ S cm}^{-1}$. To study the LMMS content effect on fiber microstructure, we added SEM images of the fibers to observe their surface, cross-section, and longitudinal-section morphologies (**Fig. R18**). It is observed that, as the LMMS content increases from 44 wt% to 66 wt%, more LMMS can be observed both on the surface and inside of the fibers. Too few LMMSs tend to be unevenly distributed and cannot form effective connections to the AgFKs. Too many LMMSs easily agglomerate due to their high surface tension, and some excess LMMSs are exposed on the fiber’s surface. Both are not good for sufficient electrical connection in the fiber, resulting in relatively poor conductivity (**Fig. R17**). The related data and clarification have been supplemented in **Supplementary Figures 6 and 7** and the manuscript (**page 6**).

Fig. R17. Conductivity of PUAL fibers with (a) different mass ratios of LMMSs to AgFKs and (b) different contents of LMMSs. Five parallel samples were measured, and error bars represent the standard deviation of the mean.

Fig. R18. SEM images of PUAL fibers prepared with different contents of LMMSs. (a) 44 wt%. (b) 54 wt%. (c) 60 wt%. (d) 66 wt%. The main LMMSs in the fibers have been marked in bright yellow.

Comment 1.7: This manuscript attributes conductivity enhancement to lubricant removal, LMMS rupture, and AgFK alignment. To clarify their relative contributions, the authors should provide experimental evidence isolating each mechanism, such as tests with and without LMMSs or heating with unaligned AgFKs, and quantify the extent to which each mechanism contributes to the overall enhancement under different conditions (e.g., high vs. low temperatures).

Response: Thank you for your insightful comment and suggestion. In the manuscript, by comparing the properties of PUA and PUAL fiber, we concluded that the highly oriented AgFKs formed a primary conducting path in the fiber, and the main contribution of LMMSs is forming dynamic electrical connections between AgFKs, to improve initial conductivity and electromechanical stability of the PUAL fiber. In addition, both heating/freezing treatments and cyclic stretching can provide extra activation effects by changing the fiber's volume or rheological behaviors to affect the fillers' connection, promoting the fiber's electrical properties^{14, 15}. Therefore, the PUAL fiber's conductivity enhancement

is cooperatively promoted by AgFKs orientation, AgFKs degreasing and LMMS deformation/rupture, as well as expansion and contraction, softening and cold/thermal-stretching activation effect of TPU matrix under different temperatures, as summarized in **Fig. R19** as follows.

Fig. R19. Summary of applicability scopes of different mechanisms.

(1) The role of AgFKs

a. AgFKs forming conducting path. To clarify the contribution of AgFKs, TPU/LMMS (PUL) fiber without AgFKs was prepared, and its morphology is shown in **Fig. R20a, b**, where the LMMSs are wrapped and isolated by the TPU matrix both inside and on the surface, rendering the LMMSs cannot form continuous conductive pathways. Therefore, the fiber is nonconductive under ambient conditions (25 °C). The conductivity is only measurable when the temperature is elevated above 150 °C, which is enhanced from $\sim 9 \times 10^{-6} \text{ S cm}^{-1}$ (150 °C) to $\sim 2 \times 10^{-5} \text{ S cm}^{-1}$ (180 °C) as the temperature continuously rises, indicating that only at high temperatures can activate the LMMS to deform/rupture and make contributions on the fiber's conductivity enhancement (**Fig. R20c**). AgFKs play a key role in the PUAL fiber either at ambient conditions or high temperatures by forming a dominating conducting path.

b. AgFKs degreasing (lubricant removal) at high temperature. The degreasing process and its effect on the fiber's conductivity enhancement can be studied through morphological as well as electrical property tests. Specifically, the lubricants removal from AgFKs was evidenced using the in-situ SEM to observe an AgFK in the fiber from 25 °C to 100 °C, and 180 °C, where the AgFK shows a morphology with increasing roughness (**Figure 3d**). As demonstrated in **Comment 1.4 (Fig. R13 and 14)**, such a process is irreversible.

The change of PUAL fiber's conductivity under cyclic heating treatments (25°C–100 °C–25 °C) (**Fig. R14a, Figure 3g**) verified the AgFK lubricant removal effect on conductivity enhancement at elevated

temperature. Since the lubricant removal is irreversible, the first heating cycle shows the most significant effect on the fiber's conductivity enhancement, increasing from $\sim 1070 \text{ S cm}^{-1}$ to 1361 S cm^{-1} . Thereafter, less and less residual lubricant can be gradually removed as the heating cycle increases, thus the fiber's conductivity at the end of each cycle gradually reaches a saturated value ($\sim 1470 \text{ S cm}^{-1}$) as the lubricant is sufficiently removed. From the result, the AgFKs degreasing can increase the fiber's conductivity to 137%. In this experiment, the conductivity after heating cycles is measured at the same temperature ($25 \text{ }^\circ\text{C}$), excluding the thermal expansion effect of TPU. Also, the LMMS rupture effect is eliminated since the heating temperature is never beyond $100 \text{ }^\circ\text{C}$ (as discussed above in the *AgFKs forming conducting path* section). Thus, the lubricant removal to sufficiently expose the AgFKs is significant in enhancing the electrical connections, resulting in heating-induced enhanced fiber conductivity.

c. The role of AgFKs orientation. To investigate the role of the AgFKs' orientation (alignment), cast film samples with random AgFKs orientation are prepared, to compare with the fiber sample with shear-induced AgFK orientation. The morphology and orientation calculations have been discussed in the response to Comment 1.2 (Fig. R2 and 3) for both PUAL film and fiber samples. The fiber samples with oriented AgFKs show 1238% times higher conductivity than the film samples with random AgFKs (fiber: $\sim 1070 \text{ S cm}^{-1}$, film: $\sim 88 \text{ S cm}^{-1}$), as well as higher ultimate strength (10.6 MPa) than the films (3.1 MPa) (Fig. R3d). The results indicate that oriented AgFKs with high overlap are necessary for forming effective and stable conducting paths. Besides, AgFKs' orientation can be tuned by varying the wet-spinning speed ($5\text{--}60 \text{ ml h}^{-1}$), which influences the shear force on the AgFKs, as demonstrated by hydrodynamic finite element analysis and SEM images of AgFKs in the PUAL fiber (Fig. R4 and 5 in Comment 1.2). An appropriate stress difference between vertical and parallel directions on AgFKs is crucial for rotating AgFKs and promoting high orientation, 20 ml h^{-1} was found to be an optimal wet-spinning speed that can achieve a high orientation of 84% (Fig. R6 in Comment 1.2). The high shear force and orientation render more AgFKs overlapping to increase stress transmission and electrical connection, enhancing both the fiber's mechanical properties and electrical conductivity (Fig. R6 in Comment 1.2).

The necessity of AgFK oriented in building stable conducting paths also holds at high and low temperatures, either with or without LMMS. When heating ($25\text{--}180 \text{ }^\circ\text{C}$) or cooling ($25\text{--} -30 \text{ }^\circ\text{C}$) the samples, no regular conductivity changes can be observed on the cast films of both PUA and PUAL (Fig. R21). In comparison, the fiber samples of both PUA and PUAL show smooth curves of conductivity change during heating and cooling, demonstrating the stable conducting path. It indicates that the shear-induced orientation of the fillers during wet-spinning is crucial to endow the fibers with stable and high conductivity under all temperature conditions. The related data and clarification have been supplemented in Supplementary Figures 4, 13, 26 and 28 and the manuscript (pages 6-8 and 13).

Fig. R20. Morphology and conductivity of PUL fibers (without AgFKs). **(a)** Surface SEM image. **(b)** Cross-sectional SEM images. **(c)** Conductivity variation of the fiber when heated from 150 °C to 180 °C. The main LMMSs in the fiber have been marked in bright yellow, which are wrapped and isolated by the TPU matrix both inside and on the surface, leading to discontinuous conductive paths.

Fig. R21. Conductivity variation of PUA and PUAL fibers and films **(a, b)** from 25 °C to 180 °C and **(c, d)** from 25 °C to -30 °C.

(2) The role of LMMSs

a. LMMS reinforcing conducting path.

To study the role of LMMS in the PUAL fiber, the conductivity of the fibers without LMMS (PUA) and with LMMS (PUAL) are compared, as shown in **Fig. R21b**, the PUAL fiber's initial conductivity is $\sim 1070 \text{ S cm}^{-1}$, which is 1337% times higher than PUA fiber's conductivity ($\sim 80 \text{ S cm}^{-1}$). That is because LMMS disperse among the AgFKs and connect AgFKs via the alloying effect to strengthen the conducting path^{14, 16}. The result indicates that although the AgFK is necessary for forming a primary conductive path, LMMS is necessary to highly improve the fiber's conductivity.

b. LMMS rupture to compensate conducting path.

Another important role of LMMS is the compensation of conductive path by releasing liquid metal upon rupture under temperature and stretching stimulations. The rupture and LM release behavior of LMMS can be observed from in-situ SEM and EDS images (**Fig. R22a**). The PUAL fiber was heated from 25 °C to 180 °C during the in-situ SEM, revealing that the LMMSs deform as the temperature increases, and the EDS images present the deformation/rupture and LM release of the LMMSs. It is consistent with the conclusion in the original manuscript that LM can rupture to compensate the conductive paths under stimulation.

As discussed in the first section (*AgFKs forming conducting path*) in this response, the LMMS rupture behavior is only not negligible at temperatures higher than 150 °C. Therefore, to further confirm the contributions of the LMMSs on the conductivity enhancement of the fiber, we compared the conductivity variations of the fiber samples with (PUAL) and without LMMSs (PUA) during the heating process, as shown in **Fig. R21b**. The conductivity of the PUA fiber has only one ankle point, where the sudden increase of the conductivity is due to the removal of AgFK's lubricant at elevated temperature. In contrast, PUAL fiber has 2 ankle points, showing an additional high slope that keeps increasing at the temperature approaching 180 °C. The comparison indicates that this additional conductivity increase is from the high-temperature induced rupture of LMMS to release LM and increase the connecting area of AgFKs to compensate the conductive paths.

Additionally, the LMMS rupture behavior in the stretching processes (high temperature, low temperature, room temperature) is discussed in the original manuscript in detail, as demonstrated in **Fig. R22b, Figures 4, 5 and Supplementary Figures 20-22 and 30**, indicating that LMMS plays an indispensable role in compensating the electrical paths of AgFKs during cyclic stretching at different temperatures. The related data and clarification have been supplemented in **Supplementary Figures 20-22 and 30** and the manuscript (**pages 10, 11 and 14**).

Fig. R22. (a) SEM images and EDS mapping of LMMSs in PUAL fiber at different temperatures (25 °C, 100 °C and 180 °C), the main deformed & ruptured LMMSs have been marked by red circles. (b) SEM images showing the surface morphology of PUAL fiber under different temperatures and stretching conditions, indicating the rupture of the LMMSs under stretching, as marked by bright yellow.

(3) The role of TPU (temperature-induced expansion, contraction and softening of TPU matrix and thermal-stretching activation effect). In addition to the fillers' functions, the reversible volume change behaviors of the TPU matrix with varied temperatures also play a crucial role in changing the fiber's conductivity. **Fig. R23a** shows that PUAL fiber's conductivity first increases rapidly once the fiber starts to be frozen until -30 °C and then decreases to a value close to the initial conductivity as the temperature naturally returns to 25 °C. Each cycle presents similar trends, which can be evidenced by the TMA tests showing a length contraction with a dL/L_0 of about -0.18% when cyclically frozen from 25 °C to -30 °C, and the deformation is largely reversible during the natural warming process to 25 °C (**Fig. R23b**). The mechanism of the TPU's effect in static heating/freezing can be understood from the aspect of molecular dynamics. The PUAL fiber works under the rubbery state of the TPU (the temperature of -34 °C–142 °C, **Fig. R16a, d**, the response to **Comment 1.5, Figure 2b, c**), where the molecular chains can change their conformation by internal rotation under cyclic stress loading and unloading. Under static heating (without stretching), the fiber will initially expand and weaken the fillers'

connection, slightly reducing the fiber's conductivity. Besides, the rheological responsive behaviors of the TPU matrix triggered by thermal-stretching activation are crucial for promoting the electrical interactions between the conductive fillers, affecting the fiber's electromechanical stability. If heating is accompanied by a stretching stimulation, it provides a thermal-stretching activation effect. Specifically, the cyclic tensile stress can promote the orientation transformation for both the TPU chains and conductive fillers, where the AgFKs' conductive path will be highly aligned and explicit, and the LMMSs' deformation or rupture will be more sufficient, both enhancing the fiber's electrical stability under cyclic stretching (**Fig. R24a, b, Figure 4 and Supplementary Figures 20-22**)¹⁷. Under cooling or freezing with simultaneous stretching, the TPU matrix will be contracted to promote the AgFKs' overlapping and the LMMSs' deformation/rupture, inducing closer electrical connections for the fillers to also increase the fiber's electrical stability (**Fig. R24a, c, Figure 5f and Supplementary Figure 30**). The related data and clarification have been supplemented in **Supplementary Figures 20-22 and 30** and the manuscript (**pages 6, 10, 11, 14**).

Fig. R23. Conductivity variation and TMA tests of PUAL fibers at different temperatures. (a) Conductivity variation between 25 °C to -30 °C. The start and end points of each segment of the curve correspond to the fiber's conductivity at 25 °C and -30 °C, respectively, with a two-hour interval between each cycle. (b) Cyclic TMA tests reveal the multiple deformation behavior of the fiber between 25 °C to -30 °C.

Fig. R24. Resistance changes of PUAL fiber under different cyclic tensile strains at different temperatures. (a) Cyclic stretching within 60% strain at 25 °C. (b) Cyclic stretching within 180% strain at 80 °C. (c) Cyclic stretching with 70% strain from -20 °C to 0 °C. 20 cycles were tested for each condition, with the last 5~10 cycles for clear plotting.

To sum up, AgFKs degreasing, AgFKs orientation, LMMS deformation/rupture, and temperature-induced expansion, contraction and softening of TPU matrix and thermal-stretching activation effect are all important for achieving high conductivity or electromechanical stability. AgFKs function to construct conducting paths by overlapping each other, while the alignment of AgFKs under the shear and traction force during wet spinning guarantees effective and stable conducting paths that are more sensitive to temperature and stretching stimulations. Meanwhile, LMMSs are important bridges to enhance the electrical connections of AgFKs. They not only increase the fiber's initial conductivity but also improve the fiber's electrical stability under cyclic stretching through an electrical compensation effect via deformation or rupture to release LM. The TPU provides a temperature/stretching-responsive matrix (with volume or molecular chain alignment variation) that can regulate the electrical connection of AgFKs and LMMSs. Therefore, the PUAL fiber has a recipe in which each component cooperatively functions to ensure the electromechanical properties, which can function under static status, with extra heating/freezing treatments, or stretching stimulation individually or simultaneously. Through the comprehensive control experiment, we believe our data and explanation could help to better understand the mechanism and necessity of our fiber recipe.

Comment 1.8: The manuscript suggests that low-temperature contraction enhances conductivity owing to AgFK alignment and LMMS rupture, yet stretching does not produce the same effect. The authors should address the difference in filler behavior between these conditions and discuss whether the localized strain or uneven distribution explains the discrepancy.

Response: Thank you for your insightful comment. We apologize for the inaccuracy of the previous description “Such an electrical enhancement mechanism also makes sense in dynamic cyclic stretching

of the fiber at low temperatures”, and we have rephrased this paragraph. TPU matrix contraction at low temperatures with and without stretching played an important role in promoting tighter connections of conductive fillers (both AgFK and LMMS), but the improvement in electrical properties presented was not the same. Low-temperature stimulation (static) can improve the fiber’s conductivity (**Fig. R23a, Figures 5b, e**), and low-temperature-stretching activation (dynamic) can only improve the fiber’s electrical stability (higher electrical stability means the fiber can sustain a larger tensile strain but shows a lower $\Delta R/R_0$), compared to that at room temperature (**Fig. R24a, c, Figures 4c and 5f**).

As shown in **Fig. R25**, for the static case (without stretching) at low temperatures, the fiber generates reversible volume contraction and induces more sufficient connections for the conductive fillers, promoting the fiber’s conductivity. In the dynamic case (under cyclic stretching) at low temperatures, axial elongation and radial contraction work together. Radial contraction similarly promotes conductivity. However, cyclic stretching induces larger tensile strains in the axial direction of the fiber, which actually damages the conducting paths and reduces the fiber’s conductivity. It is a common electrical-mechanical balance challenge in the community of elastic conductive fiber.

However, our PUAL fiber provides a tricomponent-dynamic-coordinated mechanism that can tackle this challenge and improve the fiber’s electrical stability during cyclic stretching. Specifically, both the freezing-induced contraction and tensile stress can induce the radial contraction of the fiber (**Fig. R25**), promoting the fillers’ overlapping/connection and LMMSs’ deformation/rupture, compensating the damaged conductive paths and improving the fiber’s electrical stability. Therefore, an obvious resistance increase ($\Delta R/R_0$ of $\sim 400\%$) happens in the first 10 stretching cycles at 70% strain, and then a significant resistance reduction ($\Delta R/R_0$ of $\sim 200\%$) occurs in the next 10 cycles (**Fig. R24c, Figure 5f**). This is because the LMMS conductive paths already being activated in the initial 10 stretching cycles, enhancing the fiber’s conductivity and achieving the optimal electrical stability at ~ -15 °C. However, as the temperature increases and more stretching cycles are applied, the TPU matrix’s expansion and elasticity degradation leads to a relatively loose arrangement of the conductive fillers, resulting in relatively reduced electrical stability. Throughout the cyclic stretching process at low temperatures, the fiber can sustain high tensile strain and maintain better electrical stability than that at room temperature.

To sum up, stretching actually provides both positive and negative effects on the PUAL fiber’s conductive stability, but its positive function can effectively alleviate the negative effect to highly maintain the fiber’s electrical stability. The related discussion has been clarified and strengthened in **Supplementary Figure 31** and the manuscript (**pages 13 and 14**).

Fig. R25. Force analysis of PUAL fiber under low temperature and stretching conditions.

Comment 1.9: The resistance-strain relationship at 60 °C (Figure 3f) shows a decrease in the resistance with increasing strain. This could be attributed to the enhanced filler alignment due to thermal softening. The authors validated this explanation using SEM images before and after the application of strain.

Response: Thank you for your valuable comment. We would like to clarify that the original **Figure 3f** aimed to compare the electrical stability (higher electrical stability means the fiber can sustain a larger tensile strain and shows a lower $\Delta R/R_0$) of the PUAL fibers cyclically stretched (20 cycles were performed each test, the last 5 cycles was adopted for clear plotting) with different strains (60%–180%) under different temperatures (60 °C, 80 °C and 100 °C) compared to 25 °C (**Fig. R26**), based on the working mechanisms as illustrated in the Response for **Comment 1.7**. In comparison, 80 °C provides the optimal thermal-stretching activation effect in improving the fiber's electrical stability, where the fiber can sustain a larger tensile strain of 180% and shows a lower $\Delta R/R_0$ of 1.6 compared to the cases under other conditions). The reason has been discussed in the original manuscript, that compared to the insufficient softening or over-softening of the TPU matrix under 60 °C and 100 °C, the fiber can maintain higher elasticity at 80 °C, which not only ensures the fiber's mechanical properties but also maintain the fillers' electrical connection upon cyclic stretching, enabling higher electrical stability of the fiber (**Fig. R27c-e**). Besides, unlike the fibers activated at 80 °C and 100 °C that show gradually increased $\Delta R/R_0$, the fiber activated at 60 °C shows a first decreased and then increased $\Delta R/R_0$ variation. It is because the lubricant removal process from AgFKs at 60 °C is slower than the case at 80 °C, and 100 °C, enabling the fiber's initial resistance to be higher at 60 °C (**Fig. R27f, g**), which then decreases due to the increased AgFK orientation degree under the cooperative effects of AgFKs degreasing, TPU softening, and LMMS rupture to bridge AgFKs path (**Fig. R27a-c, e**). Thereafter, decreasing $\Delta R/R_0$ happens due to the tensile strain continuously increasing until 140% strain when most of AgFKs' surface lubricants are removed.

Considering **Fig. 4c, d** and **Supplementary Fig. 22** already presented a clear difference in the electrical

stability of the PUAL fiber during thermal-stretching activation at different conditions (temperature and strain), and the **original Figure 3f** could not clearly show all the variates (temperature, tensile strain, and stretching cycles) and their coupling effect, which reduces the readability and comprehensibility. Thus, the **original Figure 3f** has been removed. The related clarification has been updated in the manuscript (**pages 10-11**).

Fig. R26. Resistance changes of the PUAL fiber under different cyclic tensile strains at different temperatures. **(a)** Cyclic stretching within 60% strain at 25 °C. **(b)** Cyclic stretching within 180% strain at 80 °C. **(c)** Cyclic stretching within 200% strain at 60 °C. **(d)** Cyclic stretching within 180% strain at 100 °C. 20 cycles were tested for each condition, with the last 5~6 cycles for clear plotting.

Fig. R27. Morphology and orientation degree of PUAL fibers at different temperatures with and without stretching and SEM images of AgFKs in PUAL fiber at different temperatures. (a-d) SEM image of longitudinal sections of PUAL fibers at (a) 25 °C without stretching, (b) heating at 60 °C for one hour, (c) stretching with 140% strain for 20 cycles at 60 °C, and (d) stretching with 140% strain for 20 cycles at 80 °C. (e) Orientation degree comparison of PUAL fibers obtained under the above four conditions. (f, g) In-situ SEM images of AgFKs in the fiber at 60 °C (f) and 80 °C (g). The main AgFKs in the fibers have been marked in bright yellow.

Comment 1.10: The manuscript reports different strain values in Supplementary Figure 24 (60% and 80%) and the main text (70% strain sustainability from -20 °C to 0 °C). The authors should reconcile this discrepancy by clarifying whether the testing conditions differed, such as strain rate or temperature control.

Response: Thank you for your suggestions. The difference in strain value is because a series of strains (60%, 70%, 80%) were applied to the fiber to study the mechanism of the cold-stretching-activation process, with the same testing method. To avoid repeating a similar result, we only put the 70% strain data in the main text, while the 60% and 80% strain data were provided as supplementary information.

To perform the cold-stretching-activation process, the fiber with an effective length of 2 cm was placed on a polytetrafluoroethylene (PTFE) substrate pre-frozen with liquid nitrogen, to create a natural warming process from -20 °C to 0 °C, where the fiber's temperature was recorded by an infrared camera in real-time. A stretching stage controlled by a stepping motor (similar to a syringe pump) was employed to stretch the non-fixed end of the fiber periodically, recording the fiber resistance simultaneously. First, the temperature control process is fixed the same; Second, cyclic stretching with a strain of 60%, 70%

and 80% was applied on each sample, respectively. We controlled the testing time (t) consistently with different stretching rates by adjusting the stroke volume and syringe's diameter, as follows.

$$\pi r^2 L = V$$

$$L = vt$$

where r refers to the syringe's diameter, L refers to the lateral stretching distance, V refers to the stroke volume, v refers to the transverse stretching rate, and t refers to the stretching time. Therefore, the testing conditions (the actual stretching speed: v) and parameters for each sample in the cold-stretching-activation process are the same. Thus, the electrical variation results of the fiber under cyclic stretching with a strain of 60%, 70% and 80% are comparable and convincing. More details about the test have been added in the *Methods* section (*Electrical measurements*).

Comment 1.11: The proposed application of the fiber in high-temperature warning firefighter suits is innovative, but requires further elaboration. Please define the critical temperature threshold for determining the danger and describe the methodology used to establish it. Experimental data demonstrating how the fiber enhances firefighters' ability to perceive temperature changes compared with the baseline should also be provided.

Response: Thank you for your valuable suggestion on our application demonstration about the high-temperature perceptive firefighter suits. Considering all comments regarding this application, we have refined this high-temperature warning system. Specifically, an additional unstretched (static) PUAL fiber was added alongside the stretched fiber to isolate the temperature effect from stretching influence and quantify the temperature perception. This helps to define the voltage threshold for high-temperature hazards. This improvement enables the firefighter suit to quantitatively monitor the environmental temperature at which the firefighters are exposed, both at rest and in motion. The detailed working mechanisms and operation processes of the improved firefighter suit are illustrated below.

The high-temperature warning system consists of a main circuit and two branch circuits (**Fig. R28a**). An 8.5 V DC power source is located in the main circuit, while branch circuit 1 is a series connection of a stretched (dynamic) PUAL fiber, and a red alarming LED, which are used to qualitatively judge whether the environment is secure or not. Tributary 2 is similar to tributary 1, which consists of an unstretched (static) PUAL fiber (~50 Ω) and a resistor (R0, ~200 Ω) in series for quantitative temperature identification.

The dynamic PUAL fiber attached to the arm produces variable resistance and partial voltage as a result of stretching and high-temperature activation. Wherein, under the same magnitude of stretching, the

resistance, voltage, and resistance variation of the fiber under flame are lower compared to the case at room temperature. The temperature threshold is defined to be the lowest threshold of burn injury (50 °C for 10 minutes)¹⁸. When the internal temperature of the firefighter suit reaches 50 °C, the surface temperature is measured to be 100 °C (**Fig. R28d**). Therefore, when the PUAL fiber's temperature reaches around 100 °C, a threshold partial voltage of the fiber was determined to be 0.8 V ($\geq 0.8V$, secure; $< 0.8V$, dangerous). Meanwhile, a signal processing module connected to the fiber determines the voltage value and enables a real-time safety display through the Bluetooth APP or on the laptop, indicating whether the firefighter is in a secure or dangerous state (**Fig. R28b, c and 29**). Meanwhile, the alarming LED assigned to higher voltages on the fireproof suit becomes brighter and brighter, also indicating firefighters are approaching a high-temperature environment. Besides, in both room and high-temperature environments, the periodic voltage signals of the stretched fiber also can reflect the firefighter's motion state. Theoretically, the firefighter's movements could be determined in real-time through voltage waveform analysis, such as with the assistance of machine learning techniques.

At the same time, the static PUAL fiber approaching the high-temperature condition shows a significantly decreased resistance due to the heating-induced conductivity enhancement mechanism, which affects the voltage dispense. By recording the voltage at both ends of the fiber and the fiber's temperature in real time, a quantified relationship can be built to indicate the environmental temperature via the voltage (**Table R1**). The real-time temperature information around the firefighter can be displayed through the Bluetooth APP on a mobile phone or a laptop, reminding both the firefighters and the staff at the monitoring station to take care or determine the safety status of the firefighter for taking appropriate measures (**Fig. R28c and 29**), improving the safety in high-temperature operation.

Although this high temperature-indicating and communicating firefighter suit setup is not yet mature enough for real-world application, we believe the improved design of the alarming module by deploying the fibers' functions brings a potential for the practical application of the wireless temperature-warning firefighter suit. This demonstration well reflects the functionality and integration capability of PUAL fiber that could support different applications. The related data and clarification have been updated in **Supplementary Figure 39** and the manuscript (**Figures 6m, n, page 17**).

Fig. R28. High-temperature warning system design for temperature detection. (a) Circuit diagram of the high-temperature warning system, where GND represents the ground. $R_1 = R_2 = 1 \text{ k}\Omega$, $R_3 = R_4 = 10 \text{ k}\Omega$. (b) Digital photo of the circuit board. (c) Screenshot of the Bluetooth APP of the wireless high-temperature warning system based on PUAL fibers. (d) Temperature difference between the inner lining and the outside of the firefighter suit.

Fig. R29. PUAL fibers acting as thermomechanical responsive electrodes for a high-temperature warning firefighter suit.

Table R1. Relationship between the temperature and the dispensed voltage at both ends of the static PUAL fiber.

Temperature (°C)	$20 \leq T \leq 60$	$60 < T \leq 100$	$100 < T \leq 140$
Voltage range (V)	$V \geq 1.7$	$0.6 \leq V < 1.7$	$0.1 \leq V < 0.6$

Reviewer #2 (Remarks to the Author):

Response: We sincerely appreciate your insightful comments and constructive suggestions on improving this manuscript. We have addressed the concerns above.

Reviewer #3 (Remarks to the Author):

In this manuscript, the authors introduce a strategy for high-performance elastic conductive fibers (ECF) adaptable to extreme temperatures. The ECF, composed of thermoplastic polyurethane (TPU), silver flakes (AgFKs), and liquid metal microspheres (LMMSs), was fabricated using a wet-spinning technique. Initially, these materials are randomly distributed within the solution; however, during the wet-spinning process, shear-induced alignment leads to the formation of regularly arranged conductive pathways. As a result, the ECF demonstrates significant mechanical and electrical performance enhancement. Furthermore, the materials used exhibit stability across a broad temperature range, enabling the presented ECF to perform reliably in extreme environments from -30 °C to 180 °C, showcasing outstanding functionality. However, as a reviewer, I find several areas in the manuscript that fall short of the standards required for publication in **Nature Communications**. Thus, I will reconsider its suitability after a major revision.

Response: We sincerely appreciate your insightful comments and constructive suggestions on improving this manuscript. We have addressed the concerns as follows.

Comment 3.1: Lines 86–87 state, “The fiber’s conductivity can be significantly enhanced by improving the mass ratio of AgFKs to TPU, and not sacrifice the mechanical properties obviously (Supplementary Fig. 2).” Although the authors claim there is “obviously” no sacrifice to mechanical properties, the figure suggests an “obvious” compromise. This should be corrected, as even a seemingly minor exaggeration raises questions about the manuscript’s overall credibility.

Response: Thank you for your suggestion, and we apologize for the description that is not rigorous enough. We originally intended to convey that the mechanical properties of the fibers are only slightly sacrificed. In the revised manuscript, we have explained the data more precisely, as “The PUA fiber’s conductivity can be significantly enhanced (~230% increment) by improving the mass ratio of AgFKs to TPU from 3 to 3.5, without obvious scarification on mechanical properties (~7% decrease in ultimate strength and ~4% decrease in fracture strain). Therefore, the optimal mass ratio of AgFKs to TPU is confirmed to be 3.5” (**Supplementary Figure 5**). We have also carefully checked the manuscript to make sure that all descriptions are accurate.

Comment 3.2: The data in Fig. 2d and 2f seem insufficient. The authors demonstrate SEM images reflecting changes due to temperature; however, this does not seem adequately substantiated. Can temperature-dependent environmental SEM be captured in a single location? If feasible, additional similar images under consistent conditions would be helpful. Furthermore, although LM can be discerned in SEM images, I believe the inclusion of EDS data would clarify the findings.

Response: Thank you for your comments. Although the SEM images in Fig. 2d and 2f are found to be universal in our material system, we agree that the data presentation is not as convincing as the SEM under a consistent location. We have supplemented the in-situ SEM images of both AgFK and LMMS of the TPU/AgFK/LMMS (PUAL) fiber, as shown in **Fig. R30** and **Fig. R31**.

(1) In-situ SEM revealing the AgFKs degreasing. This was evidenced using the in-situ SEM to observe an AgFK in the fiber from 25 °C to 180 °C, where the AgFK shows a morphology with increasing roughness (**Fig. R30**). As discussed in the manuscript, this degreasing process can improve the AgFK exposure and enhance electrical connections.

(2) In-situ SEM revealing the LMMS deformation/rupture. In-situ SEM and EDS images of the PUAL fiber from 25 °C to 180 °C reveal that the LMMSs gradually occur deformation as the temperature increases, and EDS images present the deformation/rupture of the LMMSs to release LM (**Fig. R31**). As discussed in the manuscript, the LMMS's deformation/rupture behavior plays an indispensable role in compensating the AgFKs conducting path at high temperatures and stretching cycling at different temperatures. The related data and clarification have been updated in **Supplementary Figure 15** and the manuscript (**Figures 3d, f and page 8**).

Fig. R30. SEM images of AgFKs of TPU/AgFK (PUA) fibers at different temperatures (25 °C, 100 °C and 180 °C).

Fig. R31. SEM images (a) and EDS mapping (b) of LMMSs of PUAL fibers at different temperatures (25 °C, 100 °C and 180 °C).

Comment 3.3: In Fig. 3i, the authors claim that there is no “obvious resistance increase,” yet an “obvious” increase is indeed apparent. Moreover, the authors cite Fig. 3i as evidence of long-term usability. However, on closer inspection, a substantial notch appears at the top of each cycle, implying that the linearity of resistance change with strain could not be consistently maintained. This would preclude accurate strain estimation based on resistance, raising the question of how long-term usability is achievable. It seems unlikely; how do the authors plan to resolve this issue? If unresolvable, the claim of long-term usability seems unsustainable.

Response: Thank you for your valuable comments and concerns on the long-term electromechanical stability of PUAL fibers upon cyclic stretching-releasing. This is a common challenge for all elastic conductive fibers, as their ultrafine diameter leads to easily damaged conductive paths upon stretching, especially at elevated temperatures. The long-term electromechanical stability in **Figure 3i (Figure 4h)** was evaluated based on a PUAL fiber with a diameter of 350 μm, which is a very fine geometry for ECF, so it is reasonable to have increased $\Delta R/R_0$ upon long-term cyclic stretching. We will first explain the notch in the graph, and then provide the potential solution for increasing fiber’s long-term electromechanical stability.

First, we would like to clarify the notch phenomenon at each signal peak of $\Delta R/R_0$ of the fiber during long-term stretching-releasing with 60% strain at 80 °C. This notch is due to the stretching stage’s sudden acceleration-induced PUAL fiber vibration (loosen-tighten cycles) at the maximum strain, which can be evidenced by the larger notch at the larger cycle numbers (**Fig. R32a**), indicating the increased

vibration of the fiber that is further loosened by the vibration. This can be solved by tightly fixing the PUAL fiber on the stretching stage. The retest data in **Fig. R32b** with PUAL fiber tightly fixed show a natural increase tendency (the maximum value of $\Delta R/R_0$ increases from 90% to $\sim 270\%$) before and after 1000 cycles of stretching with 60% strain at 80 °C (**Fig. R32b**).

In addition, the challenge of the electromechanical stability of PUAL can be solved by regulating the fiber's diameter. As demonstrated in **Fig. R32c**, a PUAL fiber with a diameter of 1 mm can sustain 1000 stretching cycles with 60% strain at 80 °C, with only $\sim 110\%$ resistance variation rate, where this change (the maximum value of $\Delta R/R_0$) is stable (60%–110%) within the 1000 cycles of stretching. It demonstrates that increasing the fiber diameter is an effective means to improve the fiber's electromechanical stability. Therefore, in practical applications, PUAL fiber could be fabricated into different diameters to meet different requirements. The fiber could also be twisted into yarns with larger diameters, hopefully, to further improve its electromechanical reliability to increase the application adaptivity. Thanks again for your valuable comment. The related data and discussion have been supplemented in **Supplementary Figure 25** and the manuscript (**Figure 4h, page 11**).

Fig. R32. Electrical durability of PUAL fibers during 1000 cycles of stretching under 60% strain at 80°C. (a) The fiber with a diameter of 350 μm (original manuscript). (b) The fiber with a diameter of 350 μm (retest). (c) The fiber with a diameter of 1 mm.

Comment 3.4: From my understanding of Fig. 5j and 5k, after one cycle of increased conductivity due to temperature change, the base resistance value itself appears altered. Is this application truly feasible, or is it suitable only for a single use? Likewise, the application depicted in Fig. 5l has issues. The image lacks clarity, making it difficult to discern whether the light is on or off in the unstretched state at room temperature, and in the supplemental movie, the “secure state” disappears too quickly; I could only confirm this by viewing at 0.3x speed. Furthermore, the application has critical flaws:

Comment 3.4.1: If the wearer remains still, there is no indication of whether they are in a secure or dangerous state.

Comment 3.4.2: As mentioned, if resistance changes due to temperature cause the device to enter a dangerous state, even upon returning to a safe environment, the device would not differentiate states. Does this mean the user must reinstall the device after each use? While single-use may hold meaning for firefighters, the application lacks significant impact.

I appreciate the intent to explore applications for extreme conditions, but the proposed application feels contrived. I recommend identifying alternative applications.

Response: We highly appreciate your affirmation of our efforts to explore the application of PUAL fibers in extreme scenario, and your insightful comments on improving the application demonstration in this work.

First, to address your concerns on the repeatability of sensitivity and the non-intuitive user interface of the firefighter suit application, we first clarified that the heating-induced resistance decrease dominated by the continuous AgFKs degreasing in the fiber is irreversible but still applicable for cyclic temperature perception, while the freezing-induced resistance change is reversible because it is dependent on the reversible volume change of the TPU matrix induced by temperature. In addition, we have redesigned the circuit for the firefighter's suit to achieve simultaneous detection of the firefighter's secure/dangerous status and the real-time environmental temperature. The details are elaborated as follows.

1. Multi-cycle non-contact perception of hot and cold objects: Firstly, we want to explain the mechanisms of high-temperature perception, which is mainly attributed to heating-induced AgFK lubricant removal (AgFK degreasing). The removal of AgFKs lubricant is irreversible, but multi-stage removals can be achieved by multi-cycle heating-induced partial degreasing, as shown in **Fig. R33a**. Therefore, we can take advantage of the multi-stage removal feature of AgFKs lubricant to realize non-contact perception of hot objects. As shown in **Fig. R33a**, the fiber sewn on a glove first shows a tendency to increase (TPU matrix expansion-weakened the connection of AgFKs and LMMS) and then decrease (AgFK degreasing-induced strong electrical connection) in each cycle upon approaching a hot object. When moving away from a hot object, the TPU matrix contracts and compacts the AgFKs more closely. This creates an enhanced conductive path and further decreases the resistance. During this multi-cycle perception process, although the decreasing resistance baseline indicates the irreversibility of Ag degreasing, the peaks could be identified to indicate the high temperature.

On the other hand, the low-temperature perception is reversible because it relies on the TPU's reversible contraction and expansion dependent on temperature (**Fig. R33b**). This contraction causes the AgFKs to pack more closely together, creating a stronger conducting path and reduced resistance. When the temperature returns to normal, this effect reverses and results in increased resistance.

2. High-temperature warning firefighter suit application: Considering all comments regarding this application, we have refined this high-temperature warning system, to separate the motion and temperature effects on the fiber's conductivity and increase the reliability of the sensing, as well as improve the alarming program & LED light to make the danger alarm more clearly and striking. Specifically, an additional unstretched (static) PUAL fiber was added alongside the stretched (dynamic) fiber to isolate the temperature effect from the stretching influence and quantify the temperature perception. This helps to define the voltage threshold for high-temperature hazards. This improvement enables the firefighter suit to quantitatively monitor the environmental temperature at which the firefighters are exposed, both at rest and in motion. The detailed working mechanisms and operation processes of the improved firefighter suit are illustrated below.

The high-temperature warning system consists of a main circuit and two branch circuits (**Fig. R34a**). An 8.5 V DC power source is located in the main circuit, while branch circuit 1 is a series connection of a stretched (dynamic) PUAL fiber, and a red alarming LED, which are used to qualitatively judge whether the environment is secure or not. Tributary 2 is similar to tributary 1, which consists of an unstretched (static) PUAL fiber ($\sim 50 \Omega$) and a resistor (R_0 , $\sim 200 \Omega$) in series for quantitative temperature identification.

The dynamic PUAL fiber attached to the arm produces variable resistance and partial voltage as a result of stretching and high-temperature activation. Wherein, under the same magnitude of stretching, the resistance, voltage, and resistance variation of the fiber under flame are lower compared to the case at room temperature. The temperature threshold is defined to be the lowest threshold of burn injury ($50 \text{ }^\circ\text{C}$ for 10 minutes)¹⁸. When the internal temperature of the firefighter suit reaches $50 \text{ }^\circ\text{C}$, the surface temperature is measured to be $100 \text{ }^\circ\text{C}$ (**Fig. R34d**). Therefore, when the PUAL fiber's temperature reaches around $100 \text{ }^\circ\text{C}$, a threshold partial voltage of the fiber was determined to be 0.8 V ($\geq 0.8\text{V}$, secure; $< 0.8\text{V}$, dangerous). Meanwhile, a signal processing module connected to the fiber determines the voltage value and enables a real-time safety display through the Bluetooth APP or on the laptop, indicating whether the firefighter is in a secure or dangerous state (**Fig. R34b, c and 35**). Meanwhile, the alarming LED assigned to higher voltages on the fireproof suit becomes brighter and brighter, also indicating firefighters are approaching a high-temperature environment. Besides, in both room and high-temperature environments, the periodic voltage signals of the stretched fiber also can reflect the firefighter's motion state. Theoretically, the firefighter's movements could be determined in real-time through voltage waveform analysis, such as with the assistance of machine learning techniques.

At the same time, the static PUAL fiber approaching the high-temperature condition shows a significantly decreased resistance due to the heating-induced conductivity enhancement mechanism,

which affects the voltage dispense. By recording the voltage at both ends of the fiber and the fiber's temperature in real time, a quantified relationship can be built to indicate the environmental temperature via the voltage (**Table R2**). The real-time temperature information around the firefighter can be displayed through the Bluetooth APP on a mobile phone or a laptop, reminding both the firefighters and the staff at the monitoring station to take care or determine the safety status of the firefighter for taking appropriate measures (**Fig. R34c and 35**), improving the safety in high-temperature operation.

Although the high-temperature-induced conductivity change involves an irreversible process (AgFK degreasing and LMMS rupture), we believe that our fibers have potential for reusable applications. As demonstrated in **Fig. R33a**, our fiber's resistance change can maintain the same trend for the first six times they are close to and away from a 200 °C object. To achieve repetitive monitoring of high temperatures, the law of resistance change versus the number of work times can be potentially established by machine learning. As long as the algorithm model is trained through a large amount of experimental data, the microcontroller unit (MCU) can be realized to automatically identify the number of working times and thus enter the corresponding working state recognition^{19, 20}. Although this high-temperature-perceptive and communicating firefighter suit setup is not yet mature enough for real-world application, we believe the improved design of the perception module by deploying the fibers' functions brings a potential for the practical application of the wireless temperature-warning firefighter suit. This demonstration reflects the functionality and integration capability of PUAL fiber that could support different applications. The related data and clarification have been updated in **Supplementary Figure 39** and the manuscript (**Figures 6k-n, page 17**).

Fig. R33. The PUAL fiber sewn on a glove for cyclically perceiving hot and cold objects. **(a)** Cyclic resistance variation when the glove periodically approaches and moves away from an object of 200 °C. **(b)** Cyclic resistance variation when the glove periodically approaches and moves away from an object of -40 °C.

Fig. R34. High-temperature warning system design for temperature detection. (a) Circuit diagram of the high-temperature warning system, where GND represents the ground. $R_1 = R_2 = 1 \text{ k}\Omega$, $R_3 = R_4 = 10 \text{ k}\Omega$. (b) Digital photo of the circuit board. (c) Screenshot of the Bluetooth APP of the wireless high-temperature warning system based on PUAL fibers. (d) Temperature difference between the inner lining and the outside of the firefighter suit.

Fig. R35. PUAL fibers acting as thermomechanical responsive electrodes for a high-temperature warning firefighter suit.

Table R2. Relationship between the temperature and the dispensed voltage at both ends of the static PUAL fiber.

Temperature (°C)	$20 \leq T \leq 60$	$60 < T \leq 100$	$100 < T \leq 140$
Voltage range (V)	$V \geq 1.7$	$0.6 \leq V < 1.7$	$0.1 \leq V < 0.6$

Comment 3.5: Are there any issues with liquid metal leakage? If knitted or woven into clothing, leakage would be unacceptable.

Response: Thank you for your valuable comments and concerns on the leakage of liquid metal. The concern of LMMS leakage can be allayed through understanding the fiber formation mechanism based on double diffusion and phase separation in the wet spinning⁷. Specifically, the solvent in the spinning solution diffuses into the coagulation bath, while the water (coagulation bath) diffuses into the fiber, inducing phase separation to occur in the TPU matrix, which triggers the TPU molecular chains

entanglement to solidify into fibers. This process is accompanied by the elastomer matrix contraction, wrapping most of the conductive materials and with a small portion of exposure, as demonstrated by SEM images of the PUAL fiber surface at 25 °C (**Fig. R36**). This structure could prevent LMMS leakage and ensure extra electrical connection.

In this study, the leakage of LM mainly exists under the stimulation of certain mechanical pressure and high temperature as shown in **Fig. R36**, where LMMS ruptured to compensate the conductive paths of AgFKs in PUAL fiber. As observed in **Fig. R36**, under static heat stimulation or moderate mechanical stimulation, the LMMS rupture on the surface is negligible. Only under severe mechanical deformation, such as when repeatedly handling the PUAL fibers in the experiments, leakage of LM is observed on the fiber surface, however, it does not affect the fiber's practicability. We have demonstrated that the fiber can sustain 1000 stretching cycles with 60% strain at 80 °C with a resistance change rate ($\Delta R/R_0$) of ~270% (**Figure 4h**). During the sewing and weaving, no observable LM is found to contaminate the cloth or tools, and the fiber maintains stable conductivity even after 1500 minutes of machine washing (**Fig. R37, Figures 6a, c, and f**).

Encapsulation would be an effective strategy to avoid the leakage of LMMSs under mechanical stress. For example, we encapsulated PUAL fibers using 50 wt% polydimethylsiloxane (PDMS) solution. The encapsulated fiber shows a smooth surface with a uniform PDMS layer of about 10 μm (**Fig. R38**). Meanwhile, PDMS has excellent electrical insulation and weather resistance, as well as hydrophobicity to protect the fiber from external erosion, which can improve the fiber's stability⁸. The encapsulation does not sacrifice the fiber's electrical properties. The similar conductivity-enhancement behavior as the original fiber was observed, which increases from 1070 S cm^{-1} (25 °C) to 1208 S cm^{-1} (-30 °C) and 1608 S cm^{-1} (180 °C) and 2463 S cm^{-1} (240 °C) under cooling and heating conditions (**Fig. R39a, b**), respectively. In addition, the encapsulated fiber can also be stretching-activated to improve its mechanoelectrical and thermoelectrical stability, maintaining stable resistance ($\Delta R/R_0$ of ~240% at 60% strain over 1000 stretching cycles, 80 °C, and ~240% at 80% strain, ~ -15 °C) (**Fig. R39c, d**), with similar trends align with the value without encapsulation. Excellent interfacial stability was observed on the fiber's cross-section after 1000 cycles of stretching deformation (**Fig. R39e, f**), indicating this encapsulation is an acceptable strategy for improving the fiber's electromechanical stability.

It should be noted that the encapsulated fiber only allows electrical connection at its ends, which reduces the fiber's application convenience in a certain. Thus, the encapsulation can be considered in specific applications. The related data and clarification have been supplemented in **Supplementary Figures 35-37** and the manuscript (**page 16**).

Fig. R36. SEM images of PUAL fiber under different temperatures, with or without tensile strain, indicating that LMMSs mainly rupture upon mechanical or thermal stimulations. The main deformed & ruptured LMMSs under different temperatures without tensile strain have been marked by red circles, and the main ruptured LMMSs under different stretching conditions have been marked in bright yellow.

Fig. R37. Demonstration of no visible leakage of LMMSs from PUAL fiber during crocheting and sewing. (a) Hand-crochet of PUAL fiber. (b) A fabric sewn with PUAL fibers. (c) The PUAL fiber ECG electrode in medical bandages after washed for 1500 minutes. Scale bar: 2 cm.

Fig. R38. SEM images and EDS mapping of PUAL fiber encapsulated with PDMS. (a) Cross-sectional SEM and EDS mapping. (b) Enlarged cross-sectional SEM. (c) Surface SEM. The Ga and In signals at the top left of the fiber are from the previous contamination of the stub.

Fig. R39. Properties of PDMS-encapsulated PUAL fiber. (a) Conductivity variation from 25 °C to 240 °C. (b) Conductivity variation from 25 °C to -30 °C. (c) Resistance changes under cyclic stretching with 60% strain at 80 °C over 1000 stretching cycles. (d) Resistance changes under cyclic stretching with 80% strain from -20 °C to 0 °C. 20 cycles were tested for each temperature range, with the last 10 cycles for clear plotting. (e) Surface morphology of the encapsulated fiber after cyclic stretching with 60% strain at 80 °C over 1000 stretching cycles. (f) Cross-sectional SEM image of the encapsulated fiber after cyclic stretching with 60% strain at 80 °C over 1000 stretching cycles.

Comment 3.6: The manuscript presents samples repeatedly exposed to high or low temperatures only. I am curious about the conductivity change if a sample previously exposed to high temperatures were to be exposed to low temperatures or vice versa.

Response: Thank you for your insightful comments and suggestions, which are important for improving the mechanism illustration of our manuscript. To address this question, we have added the following two sets of experiments.

Test 1: Continuous resistance changes of PUAL fiber from high to low temperature: PUAL fiber was placed on a hot-plate to heat from room temperature (25 °C) to 180 °C, then cooled down naturally to 25 °C, which was immediately moved onto a pre-frozen PTFE substrate to be rapidly cooled down from 25 °C to -30 °C. The fiber's resistance change during the whole process was recorded.

Test 2: Continuous resistance changes of PUAL fiber from low to high temperature: PUAL fiber was placed on the pre-frozen PTFE substrate to rapidly cool it from room temperature (25 °C) to -30 °C, and then naturally warmed up to 25 °C, thereafter, the fiber was immediately moved onto a hot-plate and heated to 180 °C, recording the fiber's resistance change during the whole process.

As shown in **Fig. R40**, for the fiber in **Test 1** that was heated first and then frozen, its conductivity showed a slight decrease and then a sharp increase to 3023 S cm⁻¹ (25°C–180 °C, **Fig. R40a**). The slight decrease is because of the fiber's instantaneous thermal expansion to weaken the fillers' connection, while at the initial relatively low temperature, the AgFKs degreasing induced-conductivity enhancing effect did not happen. The sharp increase of the conductivity is dominated by the AgFKs degreasing effect and the LMMS' rapture at higher temperatures. From 180 °C to 25 °C, the fiber experiences a continuous high-temperature environment and finally reaches room temperature. In this cooling process, the fiber gradually contracts and AgFKs' lubricants continuously removed due to the residual heat, both enhanced fillers' connection to increase the fiber's conductivity to 4986 S cm⁻¹. Thereafter, the freezing from 25 °C to -30 °C accelerates the fiber contraction, further enhancing the fillers' connection and improving the conductivity to 5316 S cm⁻¹.

For the fiber in **Test 2** (pre-frozen and then heated) (**Fig. R40b**), its conductivity first increased from 1070 S cm⁻¹ (25 °C) to 1166 S cm⁻¹ (-30 °C) ascribed to the freezing-induced contraction (increase electrical connection) and then decreased to 1103 S cm⁻¹ (25 °C) due to warming-induced expansion (partially weaken electrical connection) of the TPU matrix. Thereafter, when the fiber was heated from 25 °C to 180 °C, the conductivity experienced a slight decrease followed by a significant increase to 3018 S cm⁻¹ (180 °C), which shows the same behavior as the first stage of **Test 1**, based on the same mechanism.

Therefore, the PUAL fibers that go through continuous heating-freezing or freezing-heating treatments show the same conductivity change trends as those of the fibers with single heating or cooling treatment. Accordingly, we concluded that the heating and cooling processes do not interfere with each other, so PUAL fibers have enormous potential to be employed across temperatures, extending the application. The related data and clarification have been supplemented in **Supplementary Figure 38** and the manuscript (**page 16**).

Fig. R40. Continuous conductivity variation when PUAL fiber is subjected to (a) a continuous heating-natural cooling-freezing process and (b) a continuous freezing-natural warming-heating process.

Comment 3.7: In Supplementary Fig. 8e, f, should the text reference 180 °C instead of 100 °C?

Response: Thank you for pointing out this issue. We apologize for the typing error. We have corrected it and carefully checked all the text to ensure they are correct. The original **Supplementary Fig. 8** has been updated to the current **Supplementary Fig. 14**.

Reviewer #4 (Remarks to the Author):

The paper “A temperature-adaptive component-dynamic-coordinated strategy for high-performance elastic conductive fibers” reported the development of high-performance elastic conductive fibers (ECFs) that exhibit a temperature-adaptive, component-dynamic-coordinated strategy for enhanced conductivity, elasticity, and stability under extreme temperature conditions. The authors address a significant challenge in the field of electronic textiles (e-textiles): achieving a balance between conductivity, elasticity, knittability, and temperature tolerance. The study proposes a tricomponent-dynamic-coordination mechanism using thermoplastic polyurethane (TPU), silver flakes (AgFKs), and liquid metal microspheres (LMMSs), which are aligned through wet-spinning. Therefore, I would recommend its publication in Nature Communications after the minor revision.

Response: We sincerely appreciate your affirmation of the novelty and importance of our work. Following your comments and suggestions, we have carefully revised the manuscript and addressed your concerns as follows.

Comment 4.1: Mechanistic discussions could be strengthened: While the paper provides a plausible explanation for the conductivity enhancement mechanisms, some claims require further quantification or validation. For example: The authors hypothesize that softening of the TPU matrix and LMMS rupture contribute to conductivity enhancement. However, quantitative data on LMMS rupture (e.g., how much liquid metal is released) and its correlation with conductivity changes could strengthen the argument. And the role of AgFK alignment in dynamic conductivity enhancement could be explored in more detail with additional analysis (e.g., SAXS analysis) or modeling.

Response: Thank you for your insightful comment and suggestion. In the manuscript, by comparing the properties of TPU/AgFK (PUA) and TPU/AgFK/LMMS (PUAL) fiber, we concluded that the highly oriented AgFKs formed a primary conducting path in the fiber, and the main contribution of LMMSs is forming dynamic electrical connections between AgFKs, improving initial conductivity and electromechanical stability of the PUAL fiber. In addition, both heating/freezing treatments and cyclic stretching can provide extra activation effects by changing the fiber’s volume or rheological behaviors to affect the fillers’ connection, promoting the fiber’s electrical properties^{14, 15}. Therefore, the PUAL fiber’s conductivity enhancement is cooperatively promoted by AgFKs orientation, AgFKs degreasing and LMMS deformation/rupture, as well as expansion and contraction, softening and cold/thermal-stretching activation effect of TPU matrix under different temperatures, as summarized in **Fig. R41** as follows. The related data and clarification have been supplemented in **Supplementary Figure 13**.

Fig. R41. Summary of applicability scopes of different mechanisms.

(1) The role of AgFKs

a. AgFKs forming conducting path. To clarify the contribution of AgFKs, TPU/LMMS (PUL) fiber without AgFKs was prepared, and its morphology is shown in **Fig. R42a, b**, where the LMMSs are wrapped and isolated by the TPU matrix both inside and on the surface, rendering the LMMSs cannot form continuous conductive pathways. Therefore, the fiber is nonconductive under ambient conditions (25 °C). The conductivity is only measurable when the temperature is elevated above 150 °C, which is enhanced from $\sim 9 \times 10^{-6} \text{ S cm}^{-1}$ (150 °C) to $\sim 2 \times 10^{-5} \text{ S cm}^{-1}$ (180 °C) as the temperature continuously rises, indicating that only at high temperatures can activate the LMMS to deform/rupture and make contributions on the fiber's conductivity enhancement (**Fig. R42c**). AgFKs play a key role in the PUAL fiber either at ambient conditions or high temperatures by forming a dominating conducting path. The related data and clarification have been supplemented in **Supplementary Figure 4** and the manuscript (**page 6**).

Fig. R42. Morphology and conductivity of PUL fibers (without AgFKs). **(a)** Surface SEM image. **(b)** Cross-sectional SEM images. **(c)** Conductivity variation of the fiber when heated from 150 °C to 180 °C. The main LMMSs in the fiber have been marked in bright yellow, which are wrapped and isolated by the TPU matrix both inside and on the surface, leading to discontinuous conductive pathways.

b. AgFKs lubricant removal (AgFKs degreasing) at high temperature. The degreasing process and its effect on the fiber's conductivity enhancement can be studied through morphological as well as electrical property tests. Specifically, the lubricants removal from AgFKs was evidenced using the in-situ SEM to observe an AgFK in the fiber from 25 °C to 100 °C, and 180 °C, where the AgFK shows a morphology with increasing roughness. As demonstrated in **Fig. R43a**, SEM images of AgFK of PUAL fiber (heated from 25 °C to 100 °C to 180 °C, and naturally cooling to 25 °C) indicated that rougher AgFK surface induced by the elevated temperature did not recover as the temperature drops, thus such a process is irreversible as reported by literature⁹.

The change of PUAL fiber's conductivity under cyclic heating treatments (25 °C–100 °C–25 °C) (**Fig. R43b, Figure 3g**) verified the AgFK lubricant removal effect on conductivity enhancement at elevated temperature. Since the lubricant removal is irreversible, the first heating cycle shows the most significant effect on the fiber's conductivity enhancement, increasing from 1070 S cm⁻¹ to 1361 S cm⁻¹. Thereafter, less and less residual lubricant can be gradually removed as the heating cycle increases, thus the fiber's conductivity at the end of each cycle gradually reaches a saturated value (~1470 S cm⁻¹) as the lubricant is sufficiently removed. From the result, the AgFKs degreasing can increase the fiber's conductivity to 137%. In this experiment, the conductivity after heating cycles is measured at the same temperature (25 °C), excluding the thermal expansion effect of TPU. Also, the LMMS rupture effect is eliminated

since the heating temperature is never beyond 100 °C (as discussed above in the *AgFKs forming conducting path* section). Thus, the lubricant removal to sufficiently expose the AgFKs is significant in enhancing the electrical connections, resulting in heating-induced enhanced fiber conductivity. The related data and clarification have been supplemented in **Supplementary Figure 15** and the manuscript (page 8)

Fig. R43. SEM images and conductivity variation of the PUAL fiber. (a) SEM images of AgFK of PUAL fibers at different temperatures (25 °C, 100 °C, 180 °C and natural cooling from 180 °C to 25 °C). (b) Cyclic conductivity variation between 25 °C to 100 °C (the start and end points of each segment of the curve correspond to the conductivity at 25 °C and 100 °C, respectively, with a two-hour interval between each cycle).

c. The role of AgFKs orientation. To investigate the role of the AgFKs' orientation (alignment), cast film samples with random AgFKs orientation are prepared, to compare with the fiber sample with shear-induced AgFK orientation. The morphology and orientation calculation are shown below.

c-1. AgFKs orientation analysis by morphology analysis: To clarify the formation mechanism and quantify the AgFKs' orientation, we cut the PUA and PUAL fibers along the axial direction, and took SEM images of their longitudinal section to compare with those of cast films with the same compositions. As shown in **Fig. R44**, in the cast films, the AgFKs are randomly embedded in the TPU matrix without obvious orientation. In comparison, all the fiber samples show a higher orientation of AgFKs in both longitudinal-section and cross-section, indicating that the shear effect during the wet-spinning is crucial for improving the AgFKs orientation. To quantify the oriented degree, about 50 AgFKs were randomly taken from the longitudinal-section SEM images to measure the angle between their long axis direction and the fiber's axial direction, where the angle less than 45 degrees was counted as oriented (**Fig. R45a**). The ratio of oriented AgFKs is calculated and compared for the representative fiber samples (PUA and

PUAL) (**Fig. R45b**). Since the film samples include significant numbers of out-of-plane oriented AgFKs, it is not suitable for this calculation. It is observed that all the fiber samples achieved high AgFK orientations over 80%. By observing the morphology of fiber and cast film, it is evidenced that wet-spinning is crucial for improving the orientation of AgFKs. This orientation is important for increasing both the mechanical and electrical properties of the samples, as evidenced in **Fig. R45c, d** where higher tensile stress and electrical conductivity were observed from the fibers instead of the films.

c-2. Effect of wet-spinning speed on AgFKs orientation of PUAL fibers: To illustrate how shearing force and wet-spinning speeds affect the orientation of AgFKs along the fiber's axial direction, we conducted a hydrodynamic finite element analysis of AgFKs in a unit cell model, by multi-field coupling of laminar flow and solid mechanics, with transient solver and moving mesh to track the AgFK movement. The simulation results show that AgFK's orientation is closely related to the stresses induced by the shear effect at the syringe funnel (**Fig. R46a**). The stresses on AgFK parallel to the flow direction are significantly lower than those in the perpendicular direction (the cloud diagrams are processed in absolute value for ease of presentation), causing AgFKs in the perpendicular direction of the flow are prone to rotate, while those in the parallel direction of the flow are relatively stable (**Fig. R46b**). Therefore, AgFKs in the perpendicular flow direction tend to orient parallel to the flow direction. In our experiment, we also demonstrated that AgFKs' orientation can be tuned by varying the wet-spinning speed (5–60 ml h⁻¹), which influences the shear force on the AgFKs, as demonstrated by SEM images of AgFKs in the PUAL fiber (**Fig. R47**). An appropriate stress difference between vertical and parallel directions on AgFKs is crucial to rotate AgFKs and promote higher orientation, and 20 ml h⁻¹ was found to be an optimal wet-spinning speed that can achieve a high AgFK orientation of 84% (**Fig. R48**). The high shear force and orientation render more AgFKs overlapping to increase stress transmission and electrical connection, enhancing both the fiber's mechanical properties and electrical conductivity (**Fig. R48**).

c-3. Orientation factor of AgFKs calculated by WAXS: WAXS is a common technique to study the orientation of materials. To further confirm the AgFK's orientation, WAXS was performed for the PUAL fiber. First, three obvious diffraction peaks corresponding to the lattice plane of (111), (200) and (220) were observed in the WAXS profiles of PUAL fiber (**Fig. R49a**)⁴. Meanwhile, a strongly angle-dependent 2D WAXS diffraction pattern was demonstrated on the PUAL fiber (**Fig. R49b**)⁵, suggesting the existence of AgFKs' orientation. Furthermore, Hermann orientation factor can be calculated according to the following equations⁶.

$$f = \frac{3\langle \cos^2 \varphi \rangle - 1}{2}$$

$$\langle \cos^2 \varphi \rangle = \frac{\int_0^{2\pi} I(\varphi) \sin \varphi \cos^2 \varphi d\varphi}{\int_0^{2\pi} I(\varphi) \sin \varphi d\varphi}$$

The value of f is between 0 (for random orientation) and 1 (for a perfectly oriented sample). $I(\varphi)$ is the 1D intensity distribution along with the azimuthal angle φ of the (111) and (200) planes of AgFKs (**Fig. R49c, d**). In this paper, the orientation factor f of PUAL fiber was calculated to be 0.25, indicating a certain degree of orientation existed in the fiber. The relatively low orientation factor is because AgFKs align well along the axial direction but have varied orientation along the radial direction. This anisotropic orientation cannot be well reflected in WAXS, so the WAXS result can be regarded as supplementary evidence, and the directly calculated orientation from the SEM images would be more intuitive and convincing to reflect the AgFKs' orientation level in fiber samples.

To sum up, the fiber samples with oriented AgFKs show 1238% times higher conductivity than the film samples with random AgFKs (fiber: $\sim 1070 \text{ S cm}^{-1}$, film: $\sim 88 \text{ S cm}^{-1}$), as well as higher ultimate strength (10.6 MPa) than the films (3.1 MPa) (**Fig. R45d**). The results indicate that the orientated AgFKs with high overlapping are necessary for forming effective and stable conducting paths. The necessity of AgFK oriented in building stable conducting paths also holds at high and low temperatures, either with or without LMMS. When heating (25–180 °C) or cooling (25– -30 °C) the samples, no regular conductivity changes can be observed on the cast films of both PUA and PUAL (**Fig. R50**). In comparison, the fiber samples of both PUA and PUAL show smooth curves of conductivity change during heating and cooling, demonstrating the stable conducting path. It indicates that the shear-induced orientation of the fillers during wet-spinning is crucial to endow the fibers with stable high conductivity at all temperature conditions. The related data and clarification have been supplemented in **Supplementary Figures 8-12, 14, 26 and 28** and the manuscript (**Fig. 2g-i, pages 6, 7 and 13**).

Fig. R44. SEM images of PUA and PUAL fibers and cast films with the same compositions. (a) PUA fiber and cast film. (b) PUAL fiber and cast film. From left to right are the axial longitudinal section,

cross-section, and in-plane cross-section of the samples. The main AgFKs in the samples are marked in bright yellow.

Fig. R45. Orientation, conductivity, and mechanical properties of PUA and PUAL fibers, and film samples. (a) Schematic diagram of the calculation method of AgFKs' orientation in fiber's longitudinal section. (b) Axial orientation of AgFKs within PUA and PUAL fibers, calculated from the analysis of SEM images. (c, d) Mechanical and electrical properties of (c) PUA and (d) PUAL fibers compared with the films with the same compositions (five parallel samples were measured, and error bars represent the standard deviation of the mean).

Fig. R46. Hydrodynamic analysis and orientation calculation of AgFKs under different wet-spinning speeds. (a) Stress diagram of AgFKs in X direction (wet-spinning flow direction) at a flow rate of 20 mL h⁻¹ in the initial state. (b) Flow-induced orienting AgFKs at a constant time scale at a flow rate of 20 mL h⁻¹.

Fig. R47. SEM images of PUAL fibers prepared with different wet-spinning speeds. **(a)** 5 ml h^{-1} . **(b)** 20 ml h^{-1} . **(c)** 40 ml h^{-1} . **(d)** 60 ml h^{-1} . The AgFKs in the enlarged images have been marked in yellow.

Fig. R48. Orientation and properties of PUAL fibers prepared with different wet-spinning speeds. **(a)** The axial orientation of AgFKs calculated by analyzing longitudinal-section SEM photos of the fibers. **(b)** Diameter uniformity (five parallel samples were measured, and error bars represent the standard deviation of the mean). **(c)** Mechanical properties (five parallel samples were measured, and error bars represent the standard deviation of the mean). **(d)** Electrical conductivity (five parallel samples were measured, and error bars represent the standard deviation of the mean).

Fig. R49. WAXS of PUAL fiber. **(a)** WAXS curves of PUAL fiber. **(b)** 2D WAXS pattern. **(c)** The azimuthal plot of scattering at (111) along the ϕ direction in the region of 0° to 360° . **(d)** The azimuthal plot of scattering at (200) along the ϕ direction in the region of 0° to 360° .

Fig. R50. Conductivity variation of PUA and PUAL fibers and films **(a, b)** from 25°C to 180°C and **(c, d)** from 25°C to -30°C .

(2) The role of LMMSs

a. LMMS reinforcing conducting path.

To study the role of LMMS in the PUAL fiber, the conductivity of the fibers without LMMS (PUA) and with LMMS (PUAL) are compared, as shown in **Fig. R45d**, the PUAL fiber's initial conductivity is $\sim 1070 \text{ S cm}^{-1}$, which is 1337% times higher than PUA fiber's conductivity ($\sim 80 \text{ S cm}^{-1}$). That is because LMMS disperse among the AgFKs and connect AgFKs via the alloying effect to strengthen the conducting path^{14, 16}. The result indicates that although the AgFK is necessary for forming a primary conductive path, LMMS is necessary to highly improve the fiber's conductivity.

b. LMMS rupture to compensate conducting path.

Another important role of LMMS is the compensation of conductive path by releasing liquid metal upon rupture under temperature and stretching stimulations. The rupture and LM release behavior of LMMS can be observed from in situ SEM and EDS images (**Fig. R51a**). The PUAL fiber was heated from 25 °C to 180 °C during the in-situ SEM, revealing that the LMMSs deform as the temperature increases, and the EDS images present the deformation/rupture and LM release of the LMMSs. It is consistent with the conclusion in the original manuscript that LM can rupture to compensate the conductive paths under stimulation.

As discussed in the first section (*AgFKs forming conducting path*) in this response, the LMMS rupture behavior is only not negligible at temperatures higher than 150 °C. Therefore, to further confirm the contributions of the LMMSs on the conductivity enhancement of the fiber, we compared the conductivity variations of the fiber samples with (PUAL) and without LMMSs (PUA) during the heating process, as shown in **Fig. R51b**. The conductivity of the PUA fiber has only one ankle point, where the sudden increase of the conductivity is due to the removal of AgFK's lubricant at elevated temperature. In contrast, PUAL fiber has 2 ankle points, showing an additional high slope that keeps increasing at the temperature approaching 180 °C. The comparison indicates that this additional conductivity increase is from the high-temperature induced rupture of LMMS to release LM and increase the connecting area of AgFKs to compensate the conductive paths.

Additionally, the LMMS rupture behavior in the stretching processes (high temperature, low temperature, room temperature) is discussed in the original manuscript in detail, as demonstrated in **Fig. R51b, Figures 4, 5 and Supplementary Figures 20-22 and 30**, indicating that LMMS plays an indispensable role in compensating the electrical paths of AgFKs during cyclic stretching at different temperatures. The related data and clarification have been supplemented in **Supplementary Figures 20-22 and 30** and the manuscript (**pages 10, 11 and 14**).

Fig. R51. (a) SEM images and EDS mapping of LMMSs in PUAL fiber at different temperatures (25 °C, 100 °C and 180 °C), the main deformed & ruptured LMMSs have been marked by red circles. (b) SEM images showing surface morphology of PUAL fiber under different temperatures and stretching conditions, indicating the rupture of the LMMSs under stretching, as marked by bright yellow.

(3) The role of TPU (temperature-induced expansion, contraction and softening of TPU matrix and thermal-stretching activation effect). In addition to the fillers' functions, the reversible volume change behaviors of the TPU matrix with varied temperatures also play a crucial role in changing fiber's conductivity. **Fig. R52a** shows that PUAL fiber's conductivity first increases rapidly once the fiber starts to be frozen until -30 °C and then decreases to a value close to the initial conductivity as the temperature naturally returns to 25 °C. Each cycle presents similar trends, which can be evidenced by the TMA tests showing a length contraction with a dL/L_0 of about -0.18% when cyclically frozen from 25 °C to -30 °C, and the deformation is largely reversible during the natural warming process to 25 °C (**Fig. R52b**). The mechanism of the TPU's effect in static heating/freezing can be understood from the aspect of molecular dynamics. The PUAL fiber works under the rubbery state of the TPU (the temperature of -34 °C–142 °C, **Figure 2b, c**), where the molecular chains can change their conformation by internal rotation under cyclic stress loading and unloading. Under static heating (without stretching), the fiber will initially expand and weaken the fillers' connection, slightly reducing the fiber's conductivity. Besides, the

rheological responsive behaviors of the TPU matrix triggered by thermal-stretching activation are crucial for promoting the electrical interactions between the conductive fillers, affecting the fiber's electromechanical stability. If heating is accompanied by a stretching stimulation, it provides a thermal-stretching activation effect. Specifically, the cyclic tensile stress can promote the orientation transformation for both the TPU chains and conductive fillers, where the AgFKs' conductive path will be highly aligned and explicit, and the LMMSs' deformation or rupture will be more sufficient, both enhancing the fiber's electrical stability under cyclic stretching (**Fig. R53a, b, Figure 4 and Supplementary Figures 20-22**)¹⁷. Under cooling or freezing with simultaneous stretching, the TPU matrix will be contracted to promote the AgFKs' overlapping and the LMMSs' deformation/rupture, inducing closer electrical connections for the fillers to also increase the fiber's electrical stability (**Fig. R53a, c, Figure 5f and Supplementary Figure 30**). The related data and clarification have been supplemented in **Supplementary Figures 20-22 and 30** and the manuscript (**pages 4, 10, 11 and 14**).

To sum up, AgFKs degreasing, AgFKs orientation, LMMS deformation/rupture, and temperature-induced expansion, contraction and softening of the TPU matrix, along with the thermal-stretching activation effect are all important for achieving high conductivity or electromechanical stability. AgFKs function to construct conducting paths by overlapping each other, while the alignment of AgFKs under the shear and traction force during wet spinning guarantees an effective and stable conducting path. Meanwhile, LMMSs are important bridges to enhance the electrical connections of AgFKs. They not only increase the fiber's initial conductivity but also improve the fiber's electrical stability under cyclic stretching through an electrical compensation effect via deformation or rupture to release LM. The TPU provides a temperature/stretching-responsive matrix (with volume or molecular chain alignment variation) that can regulate the electrical connection of AgFKs and LMMSs. Therefore, the PUAL fiber has a recipe in which each component cooperatively functions to ensure the electromechanical properties, whether under static status, thermal treatments (heating/freezing), or mechanical stretching either individually or simultaneously. Through the comprehensive control experiment, we believe our data and explanation could help to better understand the mechanism and necessity of our fiber recipe.

Fig. R52. Conductivity variation and TMA tests of PUAL fibers at different temperatures. **(a)** Conductivity variation between 25 °C to -30 °C. The start and end points of each segment of the curve correspond to the fiber's conductivity at 25 °C and -30 °C, respectively, with a two-hour interval between each cycle. **(b)** Cyclic TMA tests reveal the multiple deformation behavior of the fiber between 25 °C to -30 °C.

Fig. R53. Resistance changes of PUAL fiber under different cyclic tensile strains at different temperatures. **(a)** Cyclic stretching within 60% strain at 25 °C. **(b)** Cyclic stretching within 180% strain at 80 °C. **(c)** Cyclic stretching with 70% strain from -20 °C to 0 °C. 20 cycles were tested for each condition, with the last 5 cycles for clear plotting.

Comment 4.2: Temperature adaptivity data could be strengthened: In the paper, the authors claimed that the ECFs autonomously enhance conductivity at extreme temperature, it would significantly bolster the paper's credibility if the authors could provide additional data on the long-term stability of the fibers when subjected to continuous thermal cycling between -30 °C and 180 °C.

Response: Thank you for your valuable suggestions. To strengthen the extreme temperature adaptivity data, we added the following two sets of experiments of continuous cycling between -30 °C and 180 °C.

Test 1: Continuous resistance changes of PUAL fiber from high to low temperature: PUAL fiber was placed on a hot-plate to heat from room temperature (25 °C) to 180 °C, then cooled down naturally

to 25 °C, which was immediately moved onto a pre-frozen PTFE substrate to be rapidly cooled down from 25 °C to -30 °C. The fiber's resistance change during the whole process was recorded.

Test 2: Continuous resistance changes of PUAL fiber from low to high temperature: PUAL fiber was placed on the pre-frozen PTFE substrate to rapidly cool it from room temperature (25 °C) to -30 °C, and then naturally warmed up to 25 °C, thereafter, the fiber was immediately moved onto a hot-plate and heated to 180 °C, recording the fibers' resistance change during the whole process.

As shown in **Fig. R54**, for the fiber in **Test 1** that was heated first and then frozen, its conductivity showed a slight decrease and then a sharp increase to 3023 S cm⁻¹ (25°C–180 °C, **Fig. R54a**). The slight decrease is because of the fiber's instantaneous thermal expansion to weaken the fillers' connection, while at the initial relatively low temperature, the AgFKs degreasing induced-conductivity enhancing effect did not happen. The sharp increase of the conductivity is dominated by the AgFKs degreasing effect and the LMMS' rupture at higher temperatures. From 180 °C to 25 °C, the fiber experiences a continuous high-temperature environment and finally reaches room temperature. In this cooling process, the fiber gradually contracts and AgFKs' lubricants are continuously removed due to the residual heat, both enhanced fillers' connection to increase the fiber's conductivity to 4986 S cm⁻¹. Thereafter, the freezing from 25 °C to -30 °C accelerates the fiber contraction, further enhancing the fillers' connection and improving the conductivity to 5316 S cm⁻¹.

For the fiber in **Test 2** (pre-frozen and then heated) (**Fig. R54b**), its conductivity first increased from 1070 S cm⁻¹ (25 °C) to 1166 S cm⁻¹ (-30 °C) ascribed to the freezing-induced contraction (increase electrical connection) and then decreased to 1103 S cm⁻¹ (25 °C) due to warming-induced expansion (partially weaken electrical connection) of the TPU matrix. Thereafter, when the fiber was heated from 25 °C to 180 °C, the conductivity experienced a slight decrease followed by a significant increase to 3018 S cm⁻¹ (180 °C), which shows the same behavior with the first stage of **Test 1**, based on the same mechanism.

Therefore, the PUAL fibers that go through continuous heating-freezing or freezing-heating treatments show the same conductivity change trends as those of the fibers with single heating or cooling treatment. Accordingly, we concluded that the heating and cooling processes do not interfere with each other, so PUAL fibers have enormous potential to be employed across temperatures, extending the application. The related data and clarification have been supplemented in **Supplementary Figure 38** and the manuscript (**page 16**).

Fig. R54. Continuous conductivity variation when PUAL fiber is subjected to (a) a continuous heating-natural cooling-freezing process and (b) a continuous freezing-natural warming-heating process.

Comment 4.3: Long-term durability and environmental stability: The paper demonstrates the fibers' stability under 1000 stretching cycles at 80 °C and cyclic heating/cooling. However, it does not provide data on their long-term durability under environmental conditions such as humidity, UV exposure, or extended thermal cycling.

Response: Thank you for your valuable comments. To address your concerns, we added three sets of experiments as follows to evaluate the fiber's environmental stability and durability.

(1) Humidity resistance: we tested the electrical stability of PUAL fibers when placed in different environments (40% humidity, 80% humidity) for 15 days (conductivity was measured every 3 days).

Fig. R55a shows that after humidity exposure for a long period of time, the fiber maintains a highly stable electrical conductivity of around 1075 S cm⁻¹. This is because the TPU matrix is a hydrophobic polymer, which can well protect the conductive fillers from oxidation and erosion.

(2) UV resistance: we measured the conductivity of the fiber under UV exposure (10 W UV intensity, 105 hours in total, conductivity measured every 7 hours), finding that there is a slight increase in conductivity from 1072 S cm⁻¹ to ~1109 S cm⁻¹ (**Fig. R55b**), indicating the thermal stability of the TPU matrix that can protect the conductive fillers from oxidation or leakage. Besides, the feeble photothermal conversion effect of the fiber under UV light (~33.7 °C) (**Fig. R55c**)²¹ could cause the heating-induced AgFK degreasing effect to enhance the conductivity slightly.

(3) Cyclic heating stability: we heated the PUAL fibers from 25 °C to 100 °C one cycle every day and recorded the fiber's conductivity after each heating cycle for 7 days. **Fig. R55d** shows that PUAL fibers' conductivity shows a large increase (the first cycle: from ~1070 S cm⁻¹ (initial) to 1361 S cm⁻¹ (after 1 heating cycle), then a small increase (to ~1450 S cm⁻¹) after 2 heating cycles, and to ~1470 S cm⁻¹ after 3 heating cycles), and eventually remaining a highly stable value of ~1470 S cm⁻¹ from 4 to 7 cycle.

Since this is a static heating-induced conductivity enhancement process under 100 °C, the LMMS rupture does not function (see details in the explanation of **Fig. R42c** in **Comment 4.1**) and the reason for conductivity increase is the different degree of AgFKs degreasing under elevated temperature. Therefore, the PUAL fiber presents a stable terminating conductivity when reaching the saturation of AgFK degreasing.

(4) Electromechanical durability and improvement strategy: In the original article, we tested the mechanical durability of PUAL fiber (diameter of 0.35 mm) with 60% strain at 80°C, finding that the fiber's resistance change shows a natural increase tendency ($\Delta R/R_0$ increases from 90% to ~270%) before and after 1000 cycles of stretching with 60% strain (**Fig. R55e**, **Figure 4h**). The electromechanical stability of PUAL can be further improved by regulating the fiber' diameter. We prepared a 1mm-thick PUAL fiber and tested the electromechanical durability, which can sustain 1000 stretching cycles with 60% strain at 80 °C, with only a maximum resistance change of 110% ($\Delta R/R_0$ is stable (60% to 110%) within the 1000 cycles of stretching (**Fig. R55f**), demonstrating that increasing the fiber diameter is an effective means to improve electromechanical stability. Therefore, in practical applications, PUAL fiber with different diameters can be selected to meet different requirements. Also, the fiber could be twisted into yarns with larger diameters, hopefully to further improve its electromechanical reliability to increase the application adaptivity. The related data and discussion have been supplemented in **Supplementary Figures 17, 25 and 34** and the manuscript (**pages 8 and 11 and 15 and 16**).

Fig. R55. Electrical durability of PUAL fibers at 80°C with 60% strain, and in different extreme conditions. **(a)** The 350 μm -thick fiber's conductivity variation over 15 days under ambient conditions (40% humidity and 80% humidity) (five parallel samples were measured, and error bars represent the standard deviation of the mean). **(b)** The 350 μm -thick fiber's conductivity variation over 15 days under UV exposure (10 W, 365 nm, distance of 12 cm) (five parallel samples were measured, and error bars represent the standard deviation of the mean). **(c)** Temperature of the fiber during switching on and turning off the UV (10 W, 365 nm, distance of 12 cm). **(d)** The 350 μm -thick fiber's conductivity variation during multi-cyclic heating (heating from 25 °C to 100 °C, one cycle every day) over 7 days (five parallel samples were measured, and error bars represent the standard deviation of the mean). **(e)** The 350 μm -thick fiber sustains over 1000 stretching cycles at 80 °C with 60% strain. **(f)** The 1 mm-thick fiber sustains over 1000 stretching cycles at 80 °C with 60% strain.

Comment 4.4: Could the authors please provide experimental evidence ensuring that no leakage of liquid metal occurs, even at the temperatures where the LMMSs are reported to rupture? This information is crucial for establishing the safety and practicality of the fibers for applications in harsh environmental conditions.

Response: Thank you for your valuable comments and concerns on the leakage of liquid metal. The concern of LMMS leakage can be allayed through understanding the fiber formation mechanism based on double diffusion and phase separation in the wet spinning⁷. Specifically, the solvent in the spinning solution diffuses into the coagulation bath, while the water (coagulation bath) diffuses into the fiber,

inducing phase separation to occur in the TPU matrix, which triggers the TPU molecular chains entanglement to solidify into fibers. This process is accompanied by the elastomer matrix contraction, wrapping most of the conductive materials and with a small portion of exposure, as demonstrated by SEM images of the PUAL fiber surface at 25 °C (**Fig. R56**). This structure could prevent LMMS leakage and ensure extra electrical connection.

In this study, the leakage of LM mainly exists under the stimulation of certain mechanical pressure and high temperature as shown in **Fig. R56**, where LMMS ruptured to compensate the conductive paths of AgFKs in PUAL fiber. As observed in **Fig. R56**, under static heat stimulation or moderate mechanical stimulation, the LMMS rupture on the surface is negligible. Only under severe mechanical deformation, such as when repeatedly handling the PUAL fibers in the experiments, leakage of LM is observed on the fiber surface, however, it does not affect the fiber's practicability. We have demonstrated that the fiber can sustain 1000 stretching cycles with 60% strain at 80 °C with a resistance change rate ($\Delta R/R_0$) increment of only ~270% (**Figure 4h**). During the sewing and weaving, no observable LM is found to contaminate the cloth or tools, and the fiber maintains stable conductivity even after 1500 minutes of machine washing (**Fig. R57, Figures 6a, c, and f**).

Encapsulation would be an effective strategy to avoid the leakage of LMMSs under mechanical stress. For example, we encapsulated PUAL fibers using 50 wt% polydimethylsiloxane (PDMS) solution. The encapsulated fiber shows a smooth surface with a uniform PDMS layer of about 10 μm (**Fig. R58**). Meanwhile, PDMS has excellent electrical insulation and weather resistance, as well as hydrophobicity to protect the fiber from external erosion, which can improve the fiber's stability⁸. The encapsulation does not sacrifice the fiber's electrical properties. The similar conductivity-enhancement behavior as the original fiber was observed, which increases from 1070 S cm^{-1} (25 °C) to 1208 S cm^{-1} (-30 °C) and 1608 S cm^{-1} (180 °C) and 2463 S cm^{-1} (240 °C) under cooling and heating conditions (**Fig. R59a, b**), respectively. In addition, the encapsulated fiber can also be stretching-activated to improve its mechano-electrical and thermoelectrical stability, maintaining stable resistance ($\Delta R/R_0$ of ~240% at 60% strain over 1000 stretching cycles, 80 °C, and ~240% at 80% strain, ~ -15 °C) (**Fig. R59c, d**), with similar trends align with the value without encapsulation. Excellent interfacial stability was observed on the fiber's cross-section after 1000 cycles of stretching deformation (**Fig. R59e, f**), indicating this encapsulation is an acceptable strategy for improving the fiber's electromechanical stability.

It should be noted that the encapsulated fiber only allows electrical connection at its ends, which reduces the fiber's application convenience in a certain. Thus, the encapsulation can be considered in specific applications. The related data and clarification have been supplemented in **Supplementary Figures 35-37** and the manuscript (**page 16**).

Fig. R56. SEM images of PUAL fiber under different temperatures, with or without tensile strain, indicating that LMMSs mainly rupture upon mechanical or thermal stimulations. The main deformed & ruptured LMMSs under different temperatures without tensile strain have been marked by red circles, and the main ruptured LMMSs under different stretching conditions have been marked in bright yellow.

Fig. R57. Demonstration of no visible leakage of LMMSs from PUAL fiber during crocheting and sewing. (a) Hand-crochet of PUAL fiber. (b) A fabric sewn with PUAL fibers. (c) The PUAL fiber ECG electrode in medical bandages after washed for 1500 minutes. Scale bar: 2 cm.

Fig. R58. SEM images and EDS mapping of PUAL fiber encapsulated with PDMS. (a) Cross-sectional SEM and EDS mapping. (b) Enlarged cross-sectional SEM. (c) Surface SEM. The Ga and In signals at the top left of the fiber are from the previous contamination of the stub.

Fig. R59. Properties of PDMS-encapsulated PUAL fiber. (a) Conductivity variation from 25 °C to 240 °C. (b) Conductivity variation from 25 °C to -30 °C. (c) Resistance changes under cyclic stretching with 60% strain at 80 °C over 1000 stretching cycles. (d) Resistance changes under cyclic stretching with 80% strain from -20 °C to 0 °C. 20 cycles were tested for each temperature range, with the last 10 cycles for clear plotting. (e) Surface morphology of the encapsulated fiber after cyclic stretching with 60% strain at 80 °C over 1000 stretching cycles. (f) Cross-sectional SEM image of the encapsulated fiber after cyclic stretching with 60% strain at 80 °C over 1000 stretching cycles.

We thank you again for all the valuable comments from the reviewers and editor, and hope that our responses and revisions could address the concerns.

References

1. Choi, C., Schlenker, E., Ha, H., Cheong, J.Y. & Hwang, B. Versatile applications of silver nanowire-based electrodes and their impacts. *Micromachines-Basel* **14**, 562 (2023).
2. Lu, Y. et al. High-performance stretchable conductive composite fibers from surface-modified silver nanowires and thermoplastic polyurethane by wet spinning. *ACS Appl. Mater. Interfaces* **10**, 2093-2104 (2018).
3. Kim, H.C., Kim, D., Lee, J.Y., Zhai, L. & Kim, J. Effect of wet spinning and stretching to enhance mechanical properties of cellulose nanofiber filament. *Int. J. Precis. Eng. Man-GT.* **6**, 567-575 (2019).
4. Wang, X. et al. Stretch-induced conductivity enhancement in highly conductive and tough hydrogels. *Adv. Mater.* **36**, 2313845 (2024).
5. Yang, J. et al. Water-induced strong isotropic MXene-bridged graphene sheets for electrochemical energy storage. *Science* **383**, 771-777 (2024).
6. Guo, H. et al. Highly anisotropic thermal conductivity of three-dimensional printed boron nitride-filled thermoplastic polyurethane composites: effects of size, orientation, viscosity, and voids. *ACS Appl. Mater. Interfaces* **14**, 14568-14578 (2022).
7. Li, C. et al. Ultra-high elongation MXene/polyurethane porous fibers with passive insulation, passive radiative heating and active heating properties for personal thermal management. *Chem. Eng. J.* **500**, 157186 (2024).
8. Zhang, Z. et al. Durable and highly sensitive flexible sensors for wearable electronic devices with PDMS-MXene/TPU composite films. *Ceram. Int.* **48**, 4977-4985 (2022).
9. Lu, D. & Wong, C.P. Thermal decomposition of silver flake lubricants. *J. Therm. Anal. Calorim.* **61**, 3-12 (2000).
10. Saiani, A. et al. Origin of multiple melting endotherms in a high hard block content polyurethane: effect of annealing temperature. *Macromolecules* **40**, 7252-7262 (2007).
11. Que, Y.-H. et al. The crystallisation, microphase separation and mechanical properties of the mixture of ether-based TPU with different ester-based TPUs. *Polymers-Basel* **13**, 3475 (2021).
12. He, M. et al. Development of high-performance thermoplastic composites based on polyurethane and ground tire rubber by in-situ synthesis. *Resour. Conserv. Recy.* **173**, 105713 (2021).
13. Cui, Z. et al. Thermoplastic polyurethane/titania/polydopamine(TPU/TiO₂/PDA) 3-D porous

- composite foam with outstanding oil/water separation performance and photocatalytic dye degradation. *Adv. Compos. Hybrid Ma.* **5**, 2801-2816 (2022).
14. Zhu, L. et al. Self-adhesive elastic conductive ink with high permeability and low diffusivity for direct printing of universal textile electronics. *ACS Nano* **18**, 34750-34762 (2024).
 15. Lu, Z. et al. Self-healing electro-optical skin for dual-mode human-machine interaction. *Nano Energy*. **135**, 110617 (2025).
 16. Zhuang, Q. et al. Wafer-patterned, permeable, and stretchable liquid metal microelectrodes for implantable bioelectronics with chronic biocompatibility. *Sci. Adv.* **9**, eadg8602 (2023).
 17. Zheng, S. et al. Pressure-stamped stretchable electronics using a nanofibre membrane containing semi-embedded liquid metal particles. *Nat. Electron.* **7**, 576-585 (2024).
 18. Yang, J., Su, Y., Song, G., Li, R. & Xiang, C. A new approach to predict heat stress and skin burn of firefighter under low-level thermal radiation. *Int. J. Therm. Sci.* **145**, 106021 (2019).
 19. Flavin, M.T. et al. Bioelastic state recovery for haptic sensory substitution. *Nature* **635**, 345-352 (2024).
 20. Liu, Z. et al. A three-dimensionally architected electronic skin mimicking human mechanosensation. *Science* **384**, 987-994 (2024).
 21. Li, X. et al. Liquid metal initiator of ring-opening polymerization: self-capsulation into thermal/photomoldable powder for multifunctional composites. *Adv. Mater.* **32**, 2003553 (2020).

Dear Reviewers:

We are sincerely grateful for your meticulous review and constructive feedback on our manuscript "A temperature-adaptive component-dynamic-coordinated strategy for high-performance elastic conductive fibers" (Research Article, NCOMMS-24-70055A). Your constructive comments have been instrumental in enhancing the depth and rigor of our study, making it more scientifically rigorous and better positioned to contribute to the scientific discourse in *Nature Communications*.

Sincerely,

Jiaqing Xiong

Professor, PhD

State Key Laboratory of Advanced Fiber Materials, College of Textiles, and Innovation Center for Textile Science and Technology, Donghua University

2999 North Renmin Road, Shanghai 201620, China

Email: jqxiong@dhu.edu.cn

REVIEWERS' COMMENTS:

Reviewer #1 (Remarks to the Author):

The authors have done a great job addressing my previous questions!

Response: We sincerely appreciate your thoughtful feedback and constructive comments. The revision process has been incredibly instructive and has helped us improve our manuscript significantly.

Reviewer #2 (Remarks to the Author):

Response: Thank you for recognizing our work and providing valuable suggestions. We have learned a great deal during the process of revising the paper.

Reviewer #3 (Remarks to the Author):

Thank you for the detailed and well-organized revision. The authors have addressed the reviewers' comments appropriately, and the revisions—particularly the enhancement of experimental data and clarification of mechanisms—contribute to a clearer presentation of the work. These updates also serve to better emphasize the novelty and relevance of the study.

The TPU-based elastic conductive fiber with highly oriented conductive fillers (AgFKs and LMMSs) exhibits a unique combination of mechanical flexibility and directional conductivity. This distinctive characteristic makes it a promising candidate for various wearable applications, including biomedical electrodes, temperature indicators, temperature-adaptive electric heaters, NFC-enabled gloves, and intelligent firefighter suits, where both durability and responsive functionality are critical.

Response: Thank you for all the valuable suggestions and recognition for our work. It is these suggestions that have enabled us to more comprehensively consider the innovations and shortcomings of this paper. We have gained a great deal during the process of revising the paper, which will also be of great inspiration for our future work.

Reviewer #4 (Remarks to the Author):

The amendments and responses submitted show an apt consideration of the reviewers' comments, enhancing the overall quality and clarity of the research. The authors have adequately addressed the proposed comments by using the latest supplementary experimental data. The revised manuscript has been improved significantly and can be accepted in Nature Communications.

Response: Thank you for your encouragement, valuable suggestions and recognition regarding our work. The comments you raised were indeed points we had not previously considered. It is precisely these suggestions that guided us to further refine the paper, making it more scientifically rigorous and better positioned to contribute to the scientific discourse in *Nature Communications*.